# Loss of *Smad4* promotes aggressive lung cancer metastasis by de-repression of PAK3 via miRNA regulation

Xiaohong Tan[1,13], Lu Tong[2,3,13], Lin Li[4,13], Jinjin Xu[3,13], Shaofang Xie[3,13], Lei Ji[3,13], Junjiang Fu[5,13], Qingwu Liu[3], Shihui Shen[3], Yun Liu[3], Yanhui Xiao[3], Feiran Gao[6,7], Robb E. Moses[8], Nabeel Bardeesy [9], Yanxiao Wang [1], Jishuai Zhang[1], Longying Tang[10], Lei Li [2], Kwok-kin Wong [11], Dianwen Song[12✉], Xiao Yang [1✉], Jian Liu [6,7✉] & Xiaotao Li [3,5,8✉]

SMAD4 is mutated in human lung cancer, but the underlying mechanism by which Smad4 loss-of-function (LOF) accelerates lung cancer metastasis is yet to be elucidated. Here, we generate a highly aggressive lung cancer mouse model bearing conditional $Kras^{G12D}$, $p53^{fl/fl}$ LOF and $Smad4^{fl/fl}$ LOF mutations (SPK), showing a much higher incidence of tumor metastases than the $Kras^{G12D}$, $p53^{fl/fl}$ (PK) mice. Molecularly, PAK3 is identified as a downstream effector of Smad4, mediating metastatic signal transduction via the PAK3-JNK-Jun pathway. Upregulation of PAK3 by Smad4 LOF in SPK mice is achieved by attenuating Smad4-dependent transcription of miR-495 and miR-543. These microRNAs (miRNAs) directly bind to the PAK3 3'UTR for blockade of PAK3 production, ultimately regulating lung cancer metastasis. An inverse correlation between Smad4 and PAK3 pathway components is observed in human lung cancer. Our study highlights the Smad4-PAK3 regulation as a point of potential therapy in metastatic lung cancer.

[1] State Key Laboratory of Proteomics, Beijing Proteome Research Center, National Center for Protein Sciences (Beijing), Beijing Institute of LifeOmics, Beijing, China. [2] Institute of Biomedical Engineering, School of Medical Instrument and Food Engineering, University of Shanghai for Science and Technology, Shanghai, China. [3] Shanghai Key Laboratory of Regulatory Biology, Institute of Biomedical Sciences, School of Life Sciences, East China Normal University, Shanghai, China. [4] Department of Orthopedic Oncology, Changzheng Hospital, The Second Military Medical University, Shanghai, China. [5] Key Laboratory of Epigenetics and Oncology, The Research Center for Preclinical Medicine, Southwest Medical University, Luzhou, Sichuan, China. [6] Zhejiang University-University of Edinburgh Institute (ZJU-UoE Institute), Zhejiang University School of Medicine, International Campus, Zhejiang University, Haining, China. [7] Department of Respiratory and Critical Care Medicine, the Second Affiliated Hospital, Zhejiang University School of Medicine, Zhejiang University, Hangzhou, China. [8] Department of Molecular and Cellular Biology, Dan L. Duncan Cancer Center, Baylor College of Medicine, One Baylor Plaza, Houston, TX, USA. [9] Cancer Center, Massachusetts General Hospital, Boston, MA, USA. [10] Shanghai Changning Maternity and Infant Health Hospital., Shanghai, China. [11] Division of Hematology & Medical Oncology, Laura and Isaac Perlmutter Cancer Center, New York University Langone Medical Center, New York, NY, USA. [12] Department of Orthopedics, Shanghai General Hospital, Shanghai Jiao Tong University, School of Medicine, Shanghai, China. [13] These authors contributed equally: Xiaohong Tan, Lu Tong, Lin Li, Jinjin Xu, Shaofang Xie, Lei Ji, Junjiang Fu. ✉email: dianwen_song@163.com; yangx@bmi.ac.cn; JianL@intl.zju.edu.cn; xiaotaol@bcm.edu

The American Cancer Society compiles cancer incidence, mortality, and deaths occurring in the United States every year. Lung cancer is the leading cause of cancer deaths worldwide, accounting for more solid tumor deaths than breast, pancreatic, prostate, and colorectal combined[1–5]. Lung cancer is broadly divided into small-cell lung cancer and non-small-cell lung cancer (NSCLC). More than 85% of lung cancers are classified as NSCLC[6]. Activating mutations of the *Kras* gene, found in 30 to 50% of NSCLC samples, are one of the most common genetic alterations in human lung cancer[7–9]. In addition, mutations of *Trp53* have been frequently reported in lung cancer (50–75%)[10]. Mutant *Kras* (hereafter called *Kras*$^{G12D}$) alone can initiate lung cancer in mice, however, the tumors rarely metastasize[11]. Cre/LoxP technology makes it possible to develop multiple conditional alleles of tumor suppressor genes or oncogenes and to initiate tumors with short latency and high penetrance[12]. In conditional lung cancer models based on nasal delivery of adenoviral CRE (*adeno-Cre*), only a subset of cells acquires mutations within the lung, mimicking the sporadic tumorigenic process[13]. Concomitant expression of *Kras*$^{G12D}$[14] and *p53*$^{fl/fl}$[15] in mouse models leads to a histologically and invasively more "humanized" version of NSCLC lung cancer[16]. Therefore, the *p53*$^{fl/fl}$; *Kras*$^{G12D}$ mouse models realistically mimic the developmental stages of human lung cancer.

*Smad4*, a tumor suppressor, is the central intracellular mediator of TGF-β signaling. Smad4 inactivation is associated with different types of cancer. For example, loss of *SMAD4* is strongly associated with increased metastatic potential and promotes pancreatic and colorectal cancer (CRC) progression[17]. Somatic mutations in the *Smad4* gene have also been described in NSCLC[18,19]. Smad4-deficient lung metastases show a significant correlation with CCL15 expression based on patient specimens[20,21]. The TGFβ-induced Smad4 complex stimulates the expression of SNAIL1 and TWIST1, which act as transcriptional factors repressing the expression of E-cadherin (EMT marker)[22]. Depletion of Smad4 also resulted in a substantial upregulation of MAPK-JNK signaling pathways[17,23]. In animal models, *Smad4* deficiency blocks TGFβ-driven epithelial to mesenchymal transition in cancer progression through multiple factors[14,24]. However, the potential mechanism of Smad4 in lung cancer metastasis has not been elucidated in vivo.

In this study, we used *adeno-Cre* to conditionally activate a *Kras*$^{G12D}$ allele with concomitant deletion of *Smad4* (*Smad4*$^{fl/fl}$) and *p53* (*p53*$^{fl/fl}$) genes to induce lung cancer. Expression of mutant *Kras*$^{G12D}$ along with *p53* and *Smad4* loss-of-function (LOF) engendered a high incidence of metastasis to different tissues, compared to that found in *p53*$^{fl/fl}$; *Kras*$^{G12D}$ mice. We found Smad4 deficiency promotes PAK3 elevation by attenuating the expression of miR-495 and miR-543, inhibitory factors for PAK3. Furthermore, activation of the PAK3-JNK-Jun pathway in the *Smad4*$^{fl/fl}$; *p53*$^{fl/fl}$; *Kras*$^{G12D}$ triple-mutant mice contributed to the metastatic potential, suggesting a possible target for therapy of NSCLC. The action of Smad4 in the regulation of PAK3 offers a tool for lung cancer prognosis. This study provides insights into Smad4-dependent regulation of tumorigenesis, progression, and metastasis in lung cancer.

## Results

### *Smad4* deletion accelerates lung tumorigenesis and metastasis.
To investigate the contribution of Smad4 LOF in lung cancer progression and metastasis in the context of conditional mouse lung cancer models, we utilized existing Cre/LoxP–controlled, genetically engineered mouse models with Kras (*Kras*$^{G12D}$)[14], p53 LOF (*p53*$^{fl/fl}$)[15], and Smad4 LOF (*Smad4*$^{fl/fl}$)[24,25] mutations to generate a cohort of lung tumors by nasal delivery of an adenovirus expressing Cre recombinase (adeno-Cre). Mice with a homozygous deletion of *Smad4* alone in lungs after adeno-Cre treatment survived beyond 52 weeks and didn't develop any gross anatomic abnormalities or lung cancer (data are not shown). Mice with *Kras*$^{G12D}$ activation (Abbreviated as *K*) alone developed lung tumors following a long latency comparable to previous reports[26,27], with no detectable metastasis (Fig. 1a). Inactivation of *Smad4* in the background of Kras$^{G12D}$ mutation (SK) had a metastasis frequency of 5.6% (Fig. 1a and Supplementary Fig. 1a). However, tumors in *Smad4*$^{fl/fl}$; *p53*$^{fl/fl}$; *Kras*$^{G12D}$ (SPK) mutant mice had a dramatically higher metastatic rate (51.2%, Fig. 1a and Supplementary Fig. 1a) than the traditional *p53*$^{fl/fl}$; *Kras*$^{G12D}$ (PK) metastatic model (13.6%, Fig. 1a, Supplementary Fig. 1a, and Supplementary Data 5). The malignant manifestation included metastatic lesions to the heart, thorax, and thymus, in addition to bone, kidney, liver, and peritoneum that are commonly observed in PK mice (Fig. 1b).

Consistent with the metastatic rate in each group, the median survival duration for SPK, PK, SK, and K mice was 12.8, 16, 26.4, and 34.6 weeks, respectively (Fig. 1c). Of note, Kaplan–Meier survival analysis showed that PK mice had prolonged median lifespan compared to SPK mice (Supplementary Fig. 1f). Most tumors exhibited features of lung cancer histologically and molecularly, as demonstrated by expression of TTF1 marker (Supplementary Fig. 1b), with higher tumor burdens in SPK than in PK mice (Supplementary Fig. 1c–e). Meanwhile, we observed less than 20% sporadic lung tumors resembling a feature of squamous cell carcinoma, such as the typical nests of neoplastic squamous cells with positive staining p63 and Krt5 (Supplementary Fig. 1g). Efficient ablation of Smad4 in SPK tumors was verified by immunohistochemical analysis of lung cancer tissues from SPK vs. PK mice (Fig. 1d) and by RT-PCR analysis of tumors for *Smad4* RNA expression (Fig. 1e). These data suggest that Smad4 suppresses tumorigenesis and metastasis of lung cancer in a mouse model.

### Abrogation of *Smad4* promotes lung cancer cell migration and invasion.
To address the action of Smad4 depletion in lung cancer cells with p53 LOF and Kras$^{G12D}$ mutation in promoting lung cancer metastasis, we isolated primary lung cancer cells from PK and SPK mouse tumors and generated immortalized PK and SPK lung cancer cell lines (Fig. 2a). Transwell assays demonstrated that SPK cells were more invasive than PK cells (Fig. 2b and Supplementary Fig. 2a). Furthermore, wound healing assays substantiated an increased cell migration in SPK cells compared to PK cells (Fig. 2c and Supplementary Fig. 2b).

Actin assembly provides a major force for cell movement by driving lamellipodia and filopodia that propel the leading edge[28]. To determine the impact of Smad4-depleted H1299 cells on the dynamic changes in lamellipodia and filopodia, stable knockdown of Smad4 by a specific shRNA (shSmad4) in the human lung cancer cell line H1299 (H1299-shSmad4, Supplementary Fig. 2c), which has an activated *RAS* and p53 LOF mutations, stimulated by serum were evaluated by cell spreading and morphological changes (Supplementary Fig. 2d). Indeed, F-actin staining revealed an enhanced formation of lamellipodia in cells silencing Smad4 (Supplementary Fig. 2e). To determine how critical Smad4 is in SPK cell invasion, we performed a "rescue experiment" by reintroducing Smad4 into SPK cells (Fig. 2d). Reexpression of Smad4 significantly attenuated SPK cell migration and invasion in a transwell study (Fig. 2e, f), while knocking down Smad4 enhanced invasion in human lung cancer cells (Supplementary Fig. 2f). Examination of the expression of EMT markers revealed that SNAIL1 was increased upon loss of Smad4 while no significant changes were observed in the expression of E-cadherin

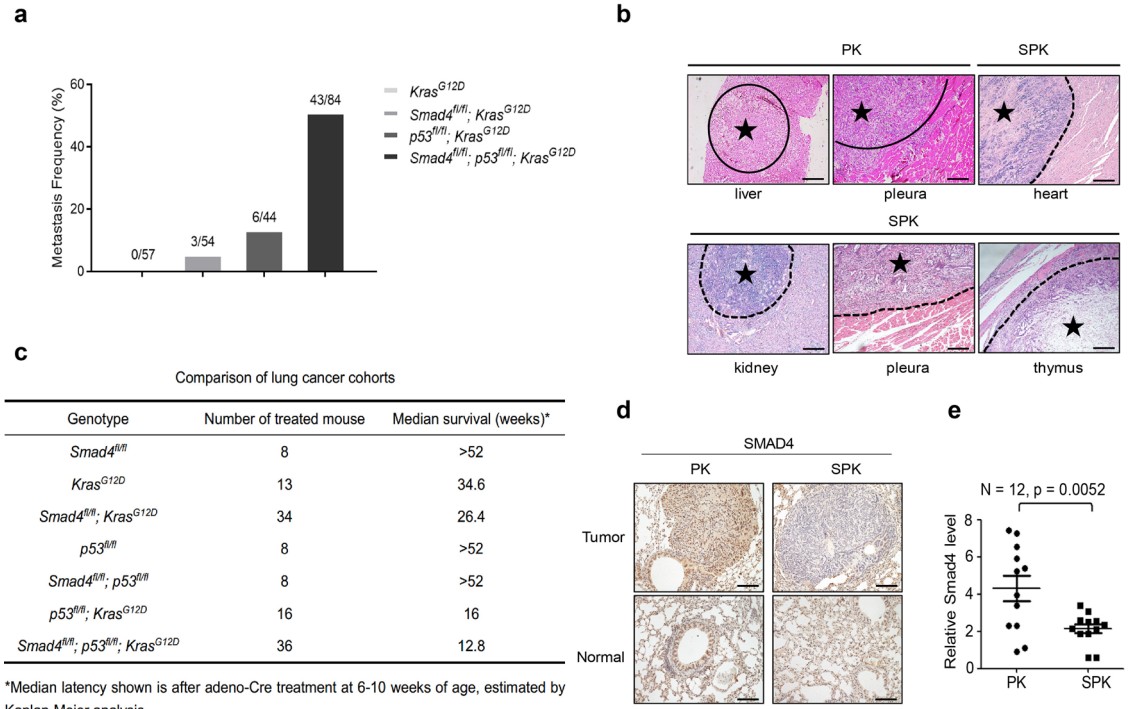

**Fig. 1 Smad4 deletion accelerates lung tumorigenesis and metastasis. a** Statistical analyses of lung cancer metastases frequency in $KRAS^{G12D}$ ($n = 57$), $Smad4^{fl/fl}$; $KRAS^{G12D}$ ($n = 54$), $p53^{fl/fl}$; $KRAS^{G12D}$ ($n = 44$), $Smad4^{fl/fl}$; $p53^{fl/fl}$; $KRAS^{G12D}$ ($n = 84$) mice. **b** Representative images of H&E stained lung cancer metastatic tissues, including liver, pleura from PK mouse and heart, kidney, pleura, thymus from SPK mouse. Similar H&E staining results were observed in a group of mice, whose information is listed in Supplementary Data 5. Asterisk indicates the area of metastatic tumors. Scale bar, 100 μm (magnification, ×10). **c** Survival statistics of different genotype mice with lung cancer. Median latency refers to the survival time on average (in weeks) after adeno-Cre treatment at the age of 6–10 weeks, estimated by Kaplan–Meier analysis. **d** Representative IHC staining of Smad4 on normal and lung tumor sections of PK or SPK mice. Mice were all treated by adeno-Cre for 14 weeks. Scale bar, 25 μm (magnification, ×40). **e** The Smad4 mRNA expression of PK and SPK mouse lung tumors was evaluated by real-time quantitative PCR. $n = 12$; data represent means ± SEM; as determined by two-tailed Student's $t$-test.

and TWIST1 between PK and SPK cells (Supplementary Fig. 2g). Overall, these results demonstrate that silencing Smad4 in the context of p53 LOF and *Kras* mutation significantly promotes migration and invasion in human and murine lung cancer cells.

**PAK3 is a downstream effector of Smad4 mediating lung cancer cell metastasis.** To understand the molecular mechanisms by which loss of function of *Smad4* promotes lung cancer cell metastasis, we compared gene expression profiles between SPK and PK model cells by RNA-sequencing (RNA-seq) analysis (Fig. 3a) which disclosed 3777 differentially expressed genes (DEGs) (Supplementary Data 1), including an increased SNAIL1 mRNA expression. We then conducted a KEGG pathway analysis on these DEGs, showing cancer as the top enriched human disease (Fig. 3b and Supplementary Data 2). Moreover, the top enriched cellular processes included cell growth and death as well as cell motility (Fig. 3b and Supplementary Data 3). To identify metastasis-associated cancer genes, we focused on genes overlapped in cancer and cell motility categories (Fig. 3c and Supplementary Data 4). Among the cancer genes involving cell motility, PAK3 was the top DEG after ablation of *Smad4* (SPK vs. PK) (Fig. 3c). We validated higher PAK3 protein and RNA expression in SPK lung tumors compared to PK tumors by immunohistochemical staining and quantitative PCR (Fig. 3d, e). Despite that Smad4 interacts with R-Smads to potentiate TGFβ signaling, our results demonstrated that regulation of PAK3 by Smad4 was not affected by the treatment of TGFβ or silencing Smad3 (Supplementary Fig. 3a), suggesting a TGFβ-independent mechanism. To substantiate the function of PAK3 is indeed correlated with SMAD4, we generated a stable knockdown of

PAK3 in an SPK cell line (Fig. 3f). In wound healing assays, PAK3 knockdown clones (shPAK3 #1 and shPAK3 #2) exhibited a reduction in wound closure compared with controls (shN) (Fig. 3g, h). Transwell assays demonstrated a much slower invasion of shPAK3 cells through the Matrigel than shN cells (Supplementary Fig. 3b, c), suggesting that PAK3 is a downstream effector of SMAD4 to promote cell metastasis. To evaluate the regulation of PAK3 in cell metastasis, we performed gain-of-function experiments using constitutively active PAK3 stably integrated in H1299 cells. Cells expressing active PAK3 (H1299-caPAK3) migrated and invaded dramatically faster than cells with vector control (H1299-vector) (Supplementary Fig. 3d–g). To assure the specific regulation of PAK3 by SMAD4, we measured the expression of PAKs (PAK1, PAK2, and PAK3) in PK and SPK cells with or without the treatment of TGFβ or TGFβ in the presence or absence of TGFβ inhibitor SB compound. We found that only PAK3, but not PAK1 or PAK2, mRNA levels are significantly upregulated upon loss of SMAD4 in a TGFβ-independent manner (Supplementary Fig. 3h). The TRI gene, a target of TGFβ signaling, was included as a positive control (Supplementary Fig. 3h). The regulation of PAK3 by SMAD4 in a TGFβ-independent manner may explain the partial changes in EMT markers. Our data suggest that the PAK3, a downstream effector of SMAD4, mediates lung cancer cell metastasis.

**PAK3 enhances the JNK-Jun signal pathway.** Given the inverse correlation between SMAD4 and PAK3 in PK/SPK lung cancer cells, we wondered how PAK3 signaling affects cell migration and invasion. In the PAK family proteins, PAK1 has been shown to

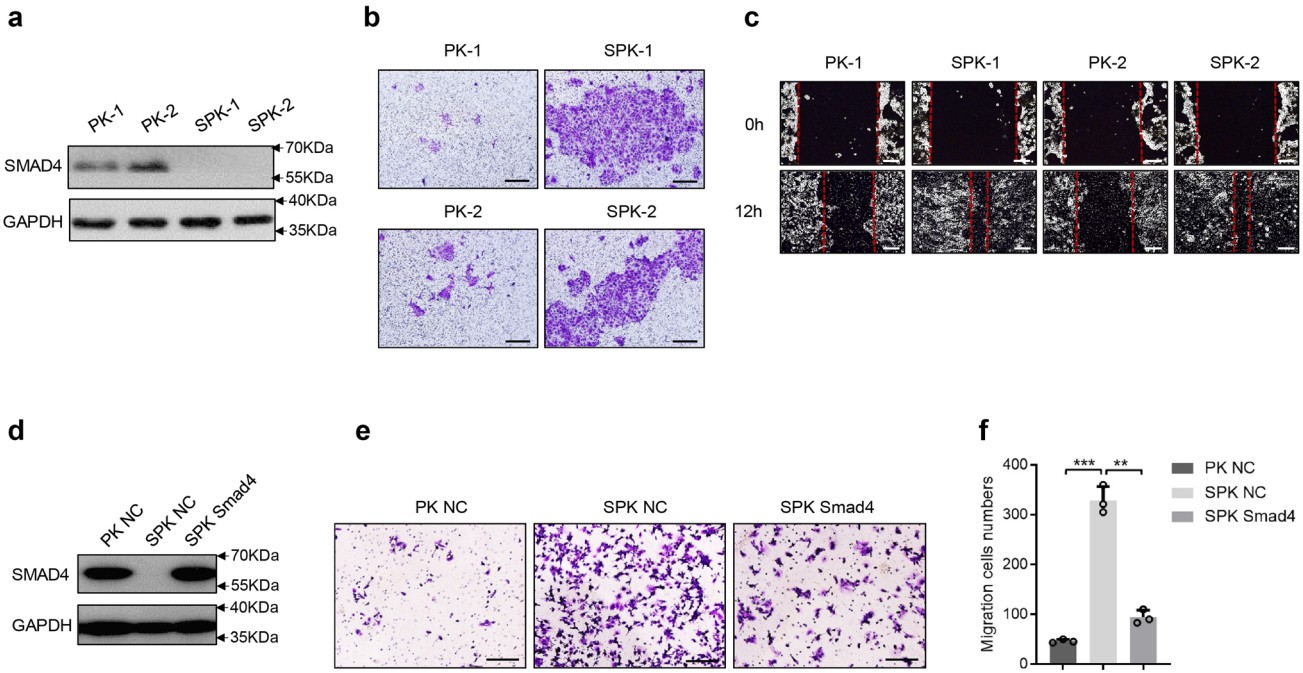

**Fig. 2 Abrogation of Smad4 promotes lung cancer cell migration and invasion. a** Primary lung cancer cells were isolated from lung tumors of PK and SPK mice and immortalized. The protein levels of Smad4 were examined in PK and SPK cells, and the experiment was repeated three times independently with similar results. Each lane represents a cell sample from an individual mouse. **b** The role of Smad4 in murine lung cancer cell migration/invasion was tested by Transwell assay and the experiment was repeated three times independently with similar results. Scale bar, 100 μm. **c** Smad4 inhibited wound healing in cell culture. Cells were made a wound between the two red dashed lines, the area of the two dashed lines represent the level of wound healing and the cell migration activity. Wound healing percentage is the ratio of the wound healing area and the primary wound area, and the experiment was repeated three times independently with similar results. Scale bar, 100 μm. **d** Transfection of Smad4 plasmid in SPK cells increased the protein level of Smad4, and the experiment was repeated three times independently with similar results. **e** Overexpression of Smad4 in SPK cells decreased cell migration/invasion, and the experiment was repeated three times independently with similar results. Scale bar, 100 μm. **f** The number of migrated cells was quantified. Data presented are the mean ± SD from three biological replicates ($n = 3$); ***$p = 6.7544E-05$; **$p = 0.000211036$; as determined by two-tailed Student's $t$-test.

promote cancer cell migration[29]. The c-Jun NH2-terminal kinase (JNK) is activated in PAK3 transfected cells, and inhibition of JNK activity abolishes PAK3-mediated cell migration in neuroendocrine tumors[30]. We also validated that blocking JNK activity abolished the motility difference between PK and SPK cells (Supplementary Fig. 4h), suggesting that JNK mediates PAK3 signaling. To determine whether PAK3 may regulate JNK-Jun activities in lung cancer cells, we checked the expression of p-JNK and p-Jun in PK and SPK cells. Lack of Smad4 and elevation of PAK3 in SPK induced significant phosphorylation of JNK with Jun (p-Jun) activation (Fig. 4a), as well as higher c-Jun mRNA levels in SPK cells than in PK cells (Supplementary Fig. 4g). Similar activation of the JNK-Jun pathway was observed in the Smad4-deficient H1299-shSmad4 cells (Supplementary Fig. 4a). In contrast, reexpression of exogenous Smad4 in SPK cells significantly suppressed the level of PAK3 and activation of the JNK-Jun kinases (Fig. 4b), suggesting that inhibition of Smad4 promotes a PAK3-dependent activation of JNK and Jun. In line with the positive regulation of PAK3 on the JNK-Jun kinases, we detected an enhanced level of P-MEK (S298), a direct downstream target of the PAK3, in SPK cells compared with PK cells (Supplementary Fig. 4i). These data demonstrate that PAK3 is activated after the loss of Smad4.

To substantiate the role of PAK3 in regulating the JNK pathway, we silenced PAK3 in H1299 cells with a panel of shRNAs and found a drastic reduction in p-JNK and p-Jun levels (Supplementary Fig. 4b). Conversely, the levels of p-JNK and p-Jun were significantly increased in H1299 cells expressing a constitutively active PAK3 (caPAK3, Supplementary Fig. 4c), suggesting that elevated PAK3 indeed activates the JNK-Jun

signal pathway. Immunohistochemical analyses substantiated positive staining of PAK3, p-JNK, and p-Jun that are negatively correlated with Smad4 in SPK lung tumor tissues compared to the PK samples (Fig. 4c, d). Therefore, our data suggest that PAK3 acts in signal transduction between Smad4 and the JNK-Jun signal pathway in lung cancer cells.

**SMAD4 negatively regulates PAK3 via transactivation of miR-495 and miR-543 expression.** As a transcription factor, SMAD4 may directly regulate PAK3 expression. Therefore, we performed bioinformatics analysis of the SMAD4 ChIP-Seq data from both human and mouse lung cells[31,32]. However, we did not observe SMAD4 binding on the PAK3 promoter region (Supplementary Fig. 4d, e). Neither did we find that exogenous expression of SMAD4, SMAD3, or SMAD2 could affect PAK3 promoter activity in PAK3 promoter-luciferase assays (Supplementary Fig. 4f). We then turned to the possibility of indirect regulation via microRNAs (miRNAs). To determine if SMAD4 regulates *PAK3* expression through its 3′ untranslated region (UTR), a known target region of miRNAs, we constructed a luciferase reporter transcriptionally fused *PAK3* 3′ UTRs downstream of the firefly luciferase gene (Fig. 5a). Interestingly, transfection of the *PAK3* 3′ UTRs luciferase reporter resulted in much higher luciferase activity in SPK than in PK cells (Fig. 5b). However, transfection of SMAD4 into SPK or H1299 cells dramatically reduced the luciferase activities (Fig. 5c and Supplementary Fig. 5a–c), leaving a potential regulatory link between SMAD4 and PAK3-targeting miRNAs. With bioinformatics analysis, we predicted that seven miRNAs may target both homo PAK3 and Mus PAK3 3′ UTRs (Supplementary Fig. 5d, indicated by stars).

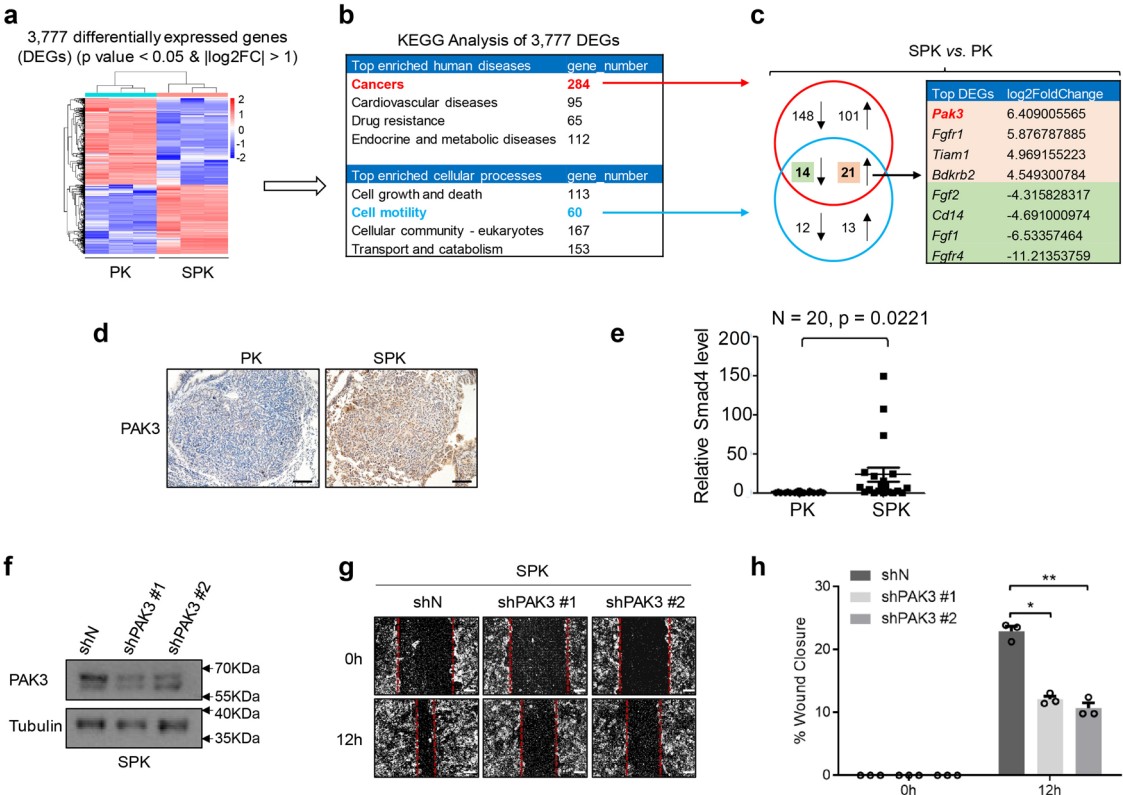

**Fig. 3 PAK3 is a downstream effector of Smad4 mediating lung cancer cell metastasis. a** Gene expression profiles were detected between PK and SPK cells by RNA-sequencing(RNA-seq) analysis. **b** KEGG Analysis of 3777 DEGs. **c** Analysis of DEGs overlapped between cancer and cell motility KEGG pathways. **d** Representative IHC images of PAK3 expression in PK and SPK mouse lung tumors. Scale bar, 50 μm (magnification, ×20). **e** Quantitative analysis of *PAK3* RNA expression in PK and SPK mouse lung tumors. Each spot represents a sample from an individual mouse. $n = 20$, data represent means ± SEM; $p < 0.001$, as determined by two-tailed Student's *t*-test. **f** Western blots showing decreased protein levels of PAK3 in shPAK3 (#1/#2) cells, and the experiment was repeated three times independently with similar results. **g** The migration ability of SPK shN and shPAK3 (#1/#2) cells were tested by wound healing assay. Cells were made a wound between the two dashed lines, the area of the two dashed lines represents the level of wound healing and the cell migration activity. Scale bar, 100 μm. **h** Statistical analyses of wound closure percentage. Data presented are the mean ± SEM from three biologically independent samples ($n = 3$); *$p = 0.000360664$; **$p = 0.000524113$; as determined by two-tailed Student's *t*-test.

Screening analysis indicated that three miRNAs, miR-495, miR-539, and miR-543, were positively regulated by SMAD4 in a dose-dependent manner in H1299 cells under TGFβ treatment (Supplementary Fig. 5e). Despite the high similarity between the homo and Mus PAK3 3′-UTR sequences (Supplementary Fig. 5g), only miR-495 and miR-543 expressions were upregulated by overexpression of SMAD4 in SPK cells (Fig. 5d). Therefore, our subsequent studies focused mainly on miR-495 and miR-543.

Consistent with the above findings, miR-495 and miR-543 levels were lower in SPK tumor samples (Fig. 5e, f). MiR-495 and miR-543 were decreased when Smad4 was transiently knocked down in H1299 cells (Supplementary Fig. 6a–c). On the contrary, these two miRNAs were increased upon overexpression of Smad4 in H1299 cells (Supplementary Fig. 6d–f). ChIP-qPCR analysis showed that SMAD4 is bound to the promoters of miR-495 and miR-543 (Fig. 5g, h and Supplementary Fig. 6g, h). Moreover, transfection of antagomir miR-495/miR-543 in PK cells exclusively led to upregulation of PAK3 mRNA without any effect on the expression of PAK1/2 (Supplementary Fig. 5f). Therefore, our data indicate that SMAD4 affects PAK3 levels by positively regulating the expression of miR-495 and miR-543.

**MiR-495 and miR-543 directly bind to the PAK3 3′UTR and attenuate the metastatic potential of lung cancer cells in vitro and in vivo.** To investigate whether miR-495 and miR-543 repress endogenous PAK3 expression, SPK cells were transfected

with oligonucleotide mimics of miR-495 or miR-543. The expression of PAK3 was significantly reduced at both mRNA and protein levels following the treatment with miR-495 or miR-543 mimics (Fig. 6a, b). Similar inhibitory effects of these two oligonucleotides on PAK3 expression were obtained in H1299 cells (Supplementary Fig. 7a, b). In contrast, transfected miR-495 or miR-543 inhibitors (antagomiR-495 or antagomiR-543) enhanced the expression of PAK3 in both PK and H1299 cells (Fig. 6c and Supplementary Fig. 7c). To determine if miR-495 and miR-543 act by directly targeting specific regions in PAK3 UTR, we generated mutant luciferase reporters with altered binding motifs of miR-495 and miR-543 in *PAK3* 3′ UTRs (Supplementary Fig. 7d). Overexpression of miR-495 or miR-543 dramatically decreased the activity of the luciferase reporter bearing the WT 3′ UTR of *PAK3* (Fig. 6d, e). However, neither miR-495 nor miR-543 inhibited the activity of the mutant luciferase reporters (Fig. 6d, e), suggesting that miR-495 and miR-543 specifically bind to the PAK3 UTR.

Then, we assessed the impact of miR-495 and miR-543 on SPK/H1299 cell migration. MiR-495 or miR-543 mimics reduced PAK3 expression and attenuated wound closure in SPK cells (Fig. 6f and Supplementary Fig. 7e) and H1299 (Supplementary Fig. 7g). An increase in cell migration/invasion was observed when antagomiR-495 or antagomiR-543 was introduced into SPK cells (Fig. 6g and Supplementary Fig. 7f) or H1299 cells (Supplementary Fig. 7h). To determine if we could block or

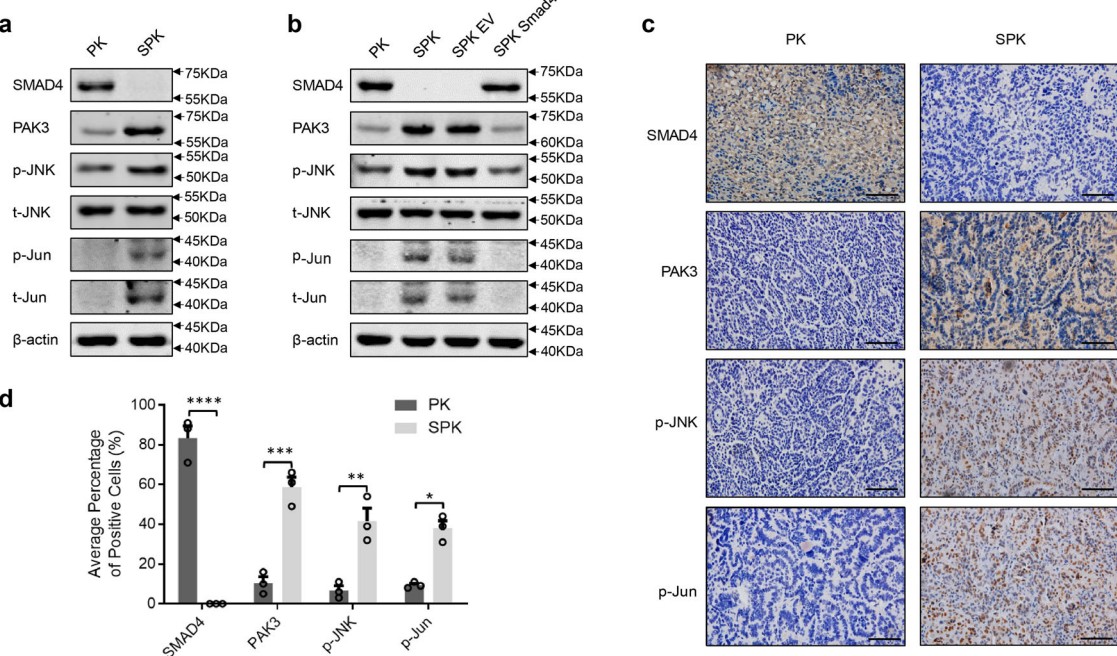

**Fig. 4 PAK3 enhances the JNK-Jun signal pathway. a** PK and SPK cells were analyzed for the expression of Smad4, PAK3, and JNK-Jun signaling-related factors by Western blotting and the experiment was repeated three times independently with similar results. **b** Western blots showing the effects of Smad4 overexpression in SPK cells and the experiment was repeated three times independently with similar results. **c** IHC staining detected the expression of Smad4, PAK3, p-JNK, and p-Jun in mouse lung tumors of PK and SPK mice. Scale bar, 25 µm (magnification, ×40). **d** Quantification of the average percentage of Smad4, PAK3, p-JNK, and p-Jun positive cells in mouse lung tumors of PK and SPK mice. Data presented are the mean ± SEM from three biologically independent samples ($n = 3$); *$p = 0.001802131$; **$p = 0.007103699$; ***$p = 0.001259358$; ****$p = 0.000180321$; as determined by two-tailed Student's $t$-test.

mimic the actions of miR-495/miR-543 with synthetic antagomirs or agomirs in mouse lung tumors, we tested miR-495/miR-543 mimics and antagomiR-495/antagomiR-543 in both male and female SPK/PK mice. To our surprise, only the miR-495 agomir can alter PAK3 expression in vivo (Fig. 6h, i), displaying downregulation of PAK3, p-JNK, and p-Jun in mouse lung tumors. In the control group, SPK mice injected with miR-NC (control) had 44.4% metastasis, in which PAK3, p-JNK, and p-Jun activities were high (Fig. 6h–j and Supplementary Fig. 8a, b). However, SPK mice injected with miR-495 mimics resulted in no tumor metastasis accompanied by reduced PAK3, p-JNK, and p-Jun levels in tumors (Fig. 6h–j). Our results demonstrate that the upregulation of PAK3 by *Smad4* LOF in SPK mice was achieved by attenuating SMAD4-dependent transcription of miRNAs that negatively regulate PAK3 expression, ultimately enhancing lung cancer metastasis.

**Correlation between SMAD4 and PAK3/JNK/Jun expression in human lung cancer samples**. Given the importance of the SMAD4-PAK3-JNK-Jun axis in experimental lung cancer metastasis, we explored the possible clinical significance of SMAD4, PAK3, p-JNK, and p-Jun expression in human lung cancers, including 15 early, 12 advanced, and 30 metastatic human lung cancer samples along with 15 normal controls (Supplementary Data 8). We found that SMAD4 was highly expressed in the controls, indicated by the highest protein levels (+++), while its expression continually dropped during the progression from early tumors to metastatic cancers. In contrast, the protein levels of PAK3, p-JNK, and p-JUN were increased in primary tumors and further in metastatic ones, compared with controls (Fig. 7a, b). By Pearson's correlation analysis on a lesion-by-lesion basis, we found a strong negative correlation between SMAD4 and PAK3 or P-JNK or P-Jun with R values of −0.29,

−0.77, and −0.73, respectively (Fig. 7c and Supplementary Fig. 10a–c). Moreover, there were highly positive correlation R values between PAK3 and P-JNK or P-Jun (Fig. 7c and Supplementary Fig. 10d, e). The above samples were also assayed for the expression of RAS (G12D) and P53 by immunohistochemistry along with organizational analysis by hematoxylin-eosin staining. The results showed that some of the metastatic samples had higher expression of RAS (G12D) and P53 (Supplementary Fig. 9a, b), reflecting enhanced RAS activity and accumulation of mutantP53 in the late stage of cancers[33,34]. Consistent with our experimental findings, bioinformatics analysis of a published human cancer dataset (between seven metastatic samples from lung cancers and 123 primary site lung tumor samples)[35] demonstrated an overall reduction in *SMAD4* expression and an elevation of *PAK3* expression displaying a strong negative correlation (Supplementary Fig. 9c, d and Data 9). These results suggested that reduced SMAD4 expression might be associated with poor prognosis in certain lung cancers. Taken together, we conclude that Smad4-mediated PAK3-JNK-Jun activation via regulation of miRNA in lung cancers appears to be a mechanism in the development of metastatic human lung cancers (Fig. 7d).

## Discussion

In our study, we have demonstrated that murine lung cancers with conditional *Smad4*$^{fl/fl}$;*p53*$^{fl/fl}$; *Kras*$^{G12D}$ mutations are a very aggressive metastatic model with a short survival duration after *adeno-Cre* delivery. We have validated the role of Smad4 LOF in promoting the invasive and metastatic progression of lung cancer in vitro and in vivo. Our results implicate the Smad4-PAK3 signal events in metastatic progression of clinical lung cancer patients.

Smad4, the central mediator of TGF-β signaling, controls the signal transduction from cell membrane to nucleus and has functions in many cellular processes, including proliferation,

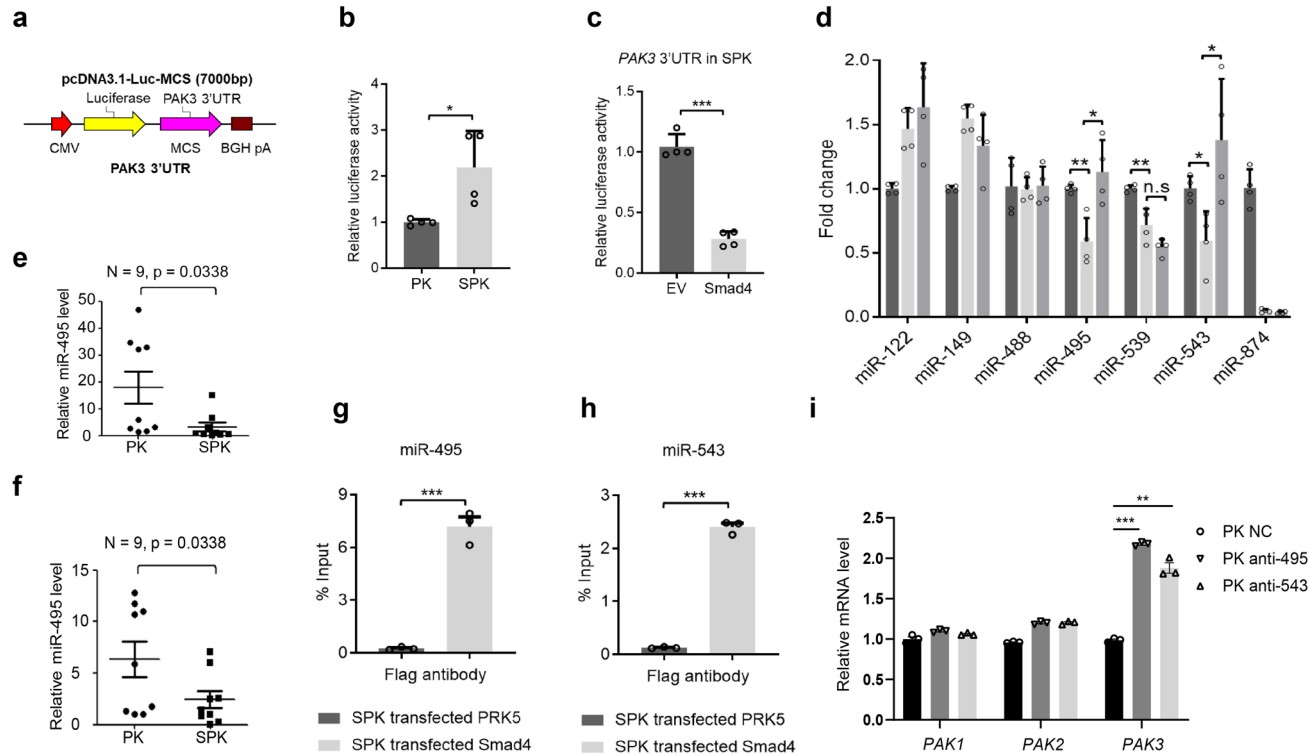

**Fig. 5 Smad4 negatively regulates PAK3 via transactivation of miR-495 and miR-543 expression. a** A luciferase reporter fused to PAK3 3′ UTRs was constructed in a pcDNA3.1-Luc-MCS plasmid. **b** Luciferase activities of PK and SPK cells transfected with PAK3 3'UTR-Luc plasmid confirmed increased PAK3 expression level SPK cells. Data presented are the mean ± SD from three biologically independent samples ($n = 3$); *$p = 0.023816011$, as determined by two-tailed Student's $t$-test. **c** Luciferase reporter assay showing that overexpression of Smad4 decreased PAK3 expression level in SPK cells. Data presented are the mean ± SEM from three biologically independent samples ($n = 3$); ***$p = 1.58613E-05$, as determined by a two-tailed Student's $t$-test. **d** RT-PCR detected expression of PAK3 correlated miRNAs in PK, SPK, and SPK overexpressing Smad4 cells. Data presented are the mean ± SEM from three biologically independent samples ($n = 3$); *$p = 0.012761254$; **$p = 0.00456168$; △$p = 0.005190084$; ○$p = 0.015974559$; ☆$p = 0.024995116$; as determined by two-tailed Student's $t$-test. **e, f** Expression levels of miR-495 and miR-543 were determined in PK and SPK mouse lung tumors. $n = 9$, data represent means ± SEM; $p < 0.05$, as determined by two-tailed Student's $t$-test. **g, h** The interaction between Smad4 and miR-495 (**g**) or miR-543 (**h**) in SPK cells were verified by ChIP assay. Data presented are the mean ± SEM from three biologically independent samples ($n = 3$); ***(**h**)$p = 6.64287E-06$; ***(**g**)$p = 0.00021899$; as determined by two-tailed Student's $t$-test. **i** qRT-PCR analysis gene expression in PK cells. Data presented are the mean ± SEM from three biologically independent samples ($n = 3$). ***$p = 5.32332E-07$; **$p = 0.000179381$.

apoptosis, and migration[36,37]. As it is deleted in most pancreatic cancer, it is also called DPC4 (deleted in pancreatic cancer)[38]. Mutations of Smad4 have been detected in pancreas cancer, colon cancer, cholangiocarcinoma cancer, and gastric cancers, suggesting an important tumor suppressor function of Smad4[39–43]. Smad4 heterozygous mice developed gastric cancer because of haploinsufficiency[44], therefore, specific Cre recombinase strategies were used to study the role of Smad4 loss in cancer development. Tissue-specific knockout of Smad4 could cause tumor formation in mammary tissue[45], including skin[46], liver[47], and colon[48]. For lung tumors, TCGA data demonstrate heterozygous Smad4 loss in 13% of lung squamous cell carcinomas and 47% of lung adenocarcinomas[49,50]. We have observed an association of reduced Smad4 expression with lung cancer malignancy in clinical metastatic samples, substantiating the role of Smad4 regulation in lung cancer progression.

Kras, p53, and Smad4 alterations are also frequently observed in other metastatic cancers, for example, in patients with metastatic CRC with 38, 60, and 27% mutation rates, respectively[51]. CRC patients with a *Kras* mutation, *p53* mutation, or *Smad4* mutation, were at a higher risk of distant metastasis[52]. A number of classical pathways show crosstalk with the *Smad4* tumor suppressor, possibly explaining why *Smad4* LOF accelerates lung cancer metastasis in combination with *Kras* and *p53* mutation. A convergence of *p53* and Smad signaling pathways has been

established[53]. Considering that some human lung tumors have not only SMAD4 mutation but also triple mutations in *KRAS*, *TP53*, and *SMAD4* (Supplementary Data 6, 7)[54], our findings may address the impact of *p53* mutation in *Smad4* LOF on the progression of these lung cancers. In addition, *Smad4* represents a barrier in *Kras*-mediated malignant transformation in a pancreatic cancer model[25,55]. In this study, our finding that Smad4 inactivates the PAK3-JNK-Jun pathway in advanced or metastatic lung cancer provides a function for the tumor-suppressive Smad4.

Several studies have indicated that TGF-β/SMAD4 signaling acts in the regulation of different miRNAs. It has been shown that TGF-β1 induces miR-574-3p transcription to inhibit cell proliferation via SMAD4 binding to the promoter of miR-574-3p[56]. In addition, the miR-23a-27a-24 cluster is induced by TGF-β1 in a Smad4-dependent manner[57]. On the other hand, miR-19b-3p promotes the proliferation of colon cancer cells by binding the 3′-UTR of *Smad4* directly[58]. In this study, we have demonstrated that miRNAs miR-495 and miR-543 are transcription targets of *Smad4*.

In summary, this study identifies an effector of Smad4, *PAK3*, and a de novo miRNA-mediated mechanism in the regulation of lung cancer metastasis. We conclude that downregulation of *Smad4* derepresses the PAK3-JNK-Jun pathway via attenuated production of miR-495/miR-543 in lung cancers. A combination of experimental, clinical, and bioinformatics analyses has

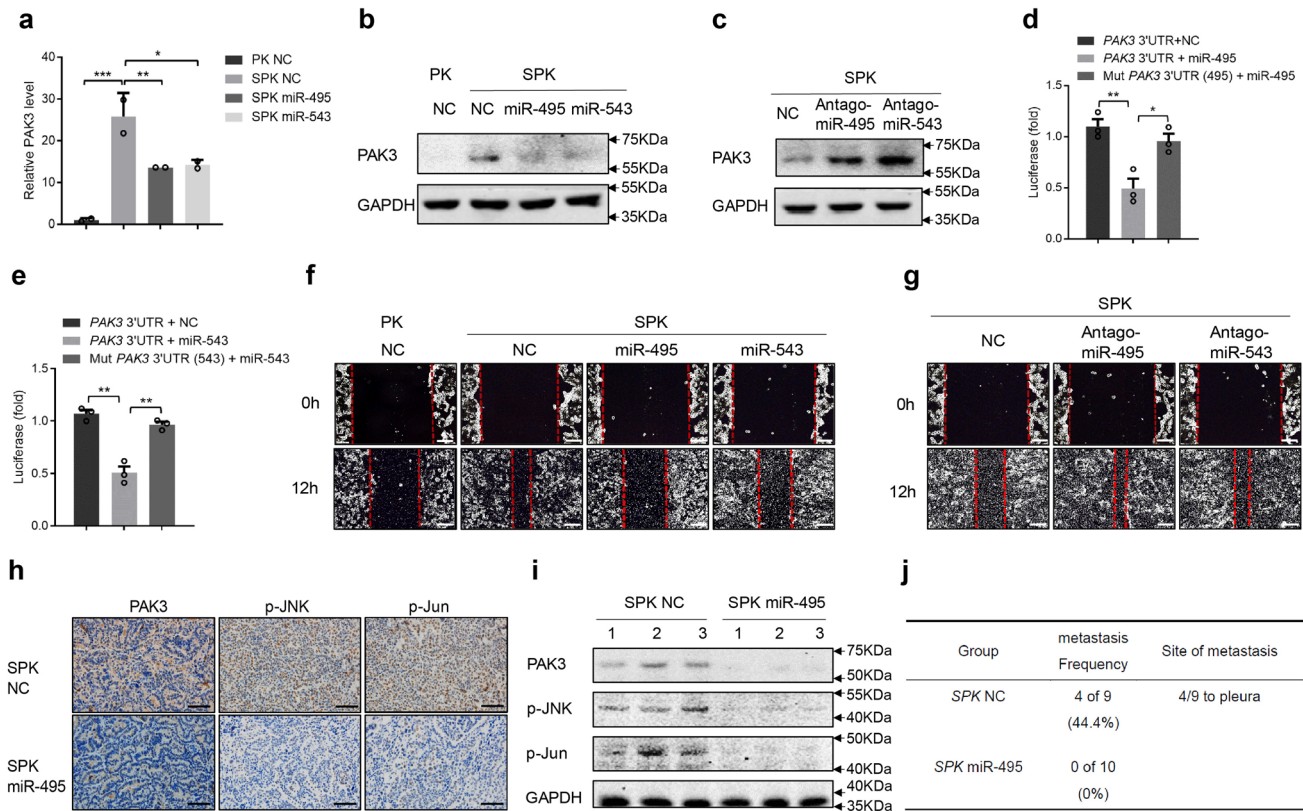

**Fig. 6 MiR-495 and miR-543 directly bind to the *PAK3* 3′ UTR and attenuate the metastatic potential of lung cancer cells in vitro and in vivo. a, b** The expression of PAK3 was detected at both mRNA and protein levels after SPK cells were transfected with miR-495 or miR-543 mimics. Data presented are the mean ± SD from three biologically independent samples ($n = 3$). ***$p = 0.000448037$; **$p = 0.007514536$; *$p = 0.009195246$. **c** The expression of PAK3 in SPK cells was detected following the transfection with miR-495 or miR-543 inhibitors (antagomiR-495 or antagomiR-543) and the experiment was repeated three times independently with similar results. **d, e** PAK3 3′UTR and its mutation luciferase report system were constructed. The activity of the luciferase reporter bearing the WT 3′ UTR of *PAK3* was measured in H1299 cells transfected with miR-495 or miR-543 mimics. Luciferase activity means the expression level of *PAK3* 3′ UTR. Data presented are the mean ± SEM from three biologically independent samples ($n = 3$); **$p = 0.007393441$ (**d**); *$p = 0.018265986$ (**d**); **$p = 0.001333882$ (**e**); *$p = 0.002263372$ (**e**); as determined by two-tailed Student's $t$-test. **f** Cells were made a wound between the two red dashed lines, the area of the two red dashed lines represent the level of wound healing and the cell migration activity. Cell migration ability was detected by wound healing assay in PK, SPK, SPK miR-495, and SPK miR-543 cells, and the experiment was repeated three times independently with similar results. Scale bar, 100 μm. **g** Cell migration ability was detected by wound healing assay after antagomiR-495 or antagomiR-543 was introduced into SPK cells, and the experiment was repeated three times independently with similar results. Scale bar, 100 μm. **h–j** Six-eight weeks SPK mice were treated by adeno-Cre and then injected NC agomir ($n = 9$) or miR-495 agomir ($n = 10$) after 4 weeks, once every 2 weeks. After 8 weeks, all mice were sacrificed to check lung cancer metastasis (**j**) and collected lung tumors to test the expression level of PAK3, p-JNK, and p-Jun, these representative mouse samples were shown by IHC (**h**), and Western blotting (**i**). Scale bar, 25 μm (magnification, ×40).

demonstrated Smad4-PAK3 regulation acts in metastatic lung cancer malignancy. Determination of Smad4/PAK3 status may be of value in stratifying patients into treatment regimens related to personalized therapy.

## Methods

**Experimental mice**. To obtain the cohorts in our study, floxed *Smad4* allele[24,59,60], *Kras*[G12D] allele[27], and *p53* conditional allele (hereafter called *p53*[fl/fl])[15] were used. PCR analyses of Kras and p53 allelic recombination were described in the previous study[61]. Smad4 primer sets were used to confirm exon deletion. The primers were listed in Supplementary Table 1. All 6–8 weeks old PK and SPK mouse were treated with $1 \times 10^6$ pfu Ad-Cre per mice by nasal drip to induce lung cancer. All experiments were conducted under specific pathogen-free (SPF) conditions and handled according to the ethical and scientific standards by the Animal Center at Shanghai Key Laboratory of Regulatory Biology, Institute of Biomedical Sciences, School of Life Sciences, East China Normal University following procedures approved by the Institutional Animal Care and Use Committee. The group size was determined based on the results of preliminary experiments[62] and no statistical method was used to predetermine sample size in animal studies. To implement random assignment, we assigned a unique number to every member of our study's sample. Then, we used a random number generator to randomly assign each number to a control or experimental group. All strains were B6. 129. To evaluate the production of metastasis in mouse models, half of the organ sample was

sectioned into slides (5-μm thickness per slide), and five representative slides were selected from each sample for the H&E staining. Once there are tumors found in organs other than lungs, immunohistochemical staining on TTF1 (a marker to demonstrate the origin of metastatic lesions from lung adenocarcinoma) would be conducted, and the tumors with the positive staining of TTF1 are considered as metastasized from the lungs.

**RNA extraction and RNA-seq**. Total RNA of SPK and SPK model cells was extracted using RNeasy kit (Qiagen, Valencia, CA) and quantified with NanoDrop 1000 (Thermo Fisher Scientific, Waltham, MA). The cDNA sequencing libraries were prepared using Illumina's TruSeq Sample Preparation Kit (San Diego, CA) and the sequencing was performed using Illumina Genome Analyzer. RNA-seq data were analyzed using R with various packages. The differential analysis of genes was conducted on counts using the DESeq2 package. DEGs were identified as such if the fold change >2 and the $p$ value <0.05. Gene ontology (GO) enrichment and enriched KEGG (Kyoto Encyclopedia of Genes and Genomes) pathways were performed.

**Cell culture**. The mouse lung cancer-derived cell lines PK and SPK were established from in vivo PK and SPK tumors. Briefly, tumors were dissected 12 weeks post Ad-Cre virus treatment, minced into small pieces, and digested with collagenase for 1 h at 37 °C. Digested tissue was filtered through a 100 um filter, then a 40 um filter using excess cold PBS to wash cell through a filter. Finally, the tumor cells were cultured in RPMI-1640 (Hyclone) with 10% FBS. The medium was

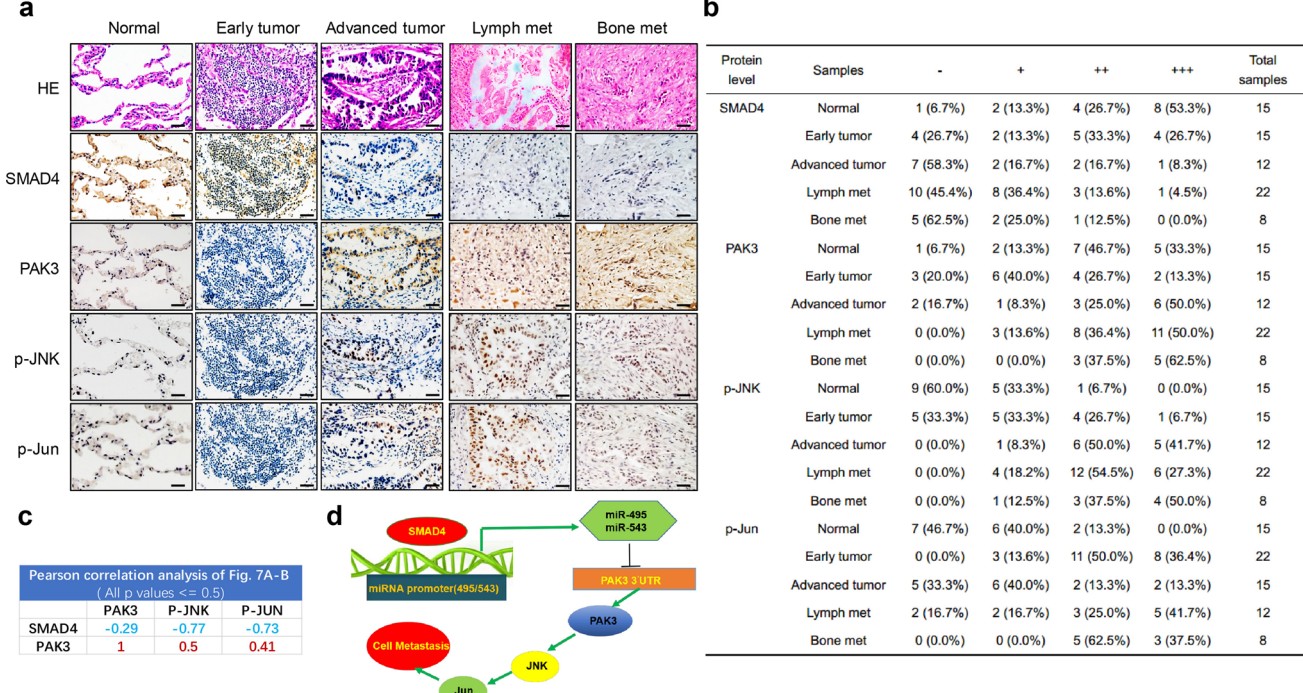

**Fig. 7 Correlation between Smad4 and PAK3/JNK/Jun expression in human lung cancer samples. a** Representative IHC images displaying an inverse correlation between Smad4, PAK3, p-JNK, and p-Jun in human normal lung tissues, lung tumors, and lung cancer metastatic samples. Scale bar, 25 μm (magnification, ×40). **b** Statistics of positively stained percentages in human normal lung tissues ($n = 15$) and lung cancer metastatic tissues: early tumor ($n = 15$); advanced tumor ($n = 12$); lymph nodes metastases ($n = 22$); bone metastases ($n = 8$). **c** Pearson correlation analysis of protein expressions in Fig. 7a, b. **d** A proposed model for the role of Smad4 in lung cancer metastasis.

changed every day until cells outgrew and stable immortalized cell lines were formed.

The human lung cancer-derived cell lines H1299 were maintained in RPMI-1640 Medium (Invitrogen) with 10% fetal bovine serum (HyClone). shN (target sequence AGCGGACTAAGTCCATTGC) and shSmad4 (target sequence GGATTTCCTCATGTGATCT) plasmids in H1299 were kindly provided by Dr. Xinhua Feng. shN and shSmad4 were transfected to H1299 cells and selected at 2 ug/ml puromycin. The pGIPZ-PAK3 (target TAGTGCTTCGTTTACTTTG), the pGIPZ-PAK3 (target TGTATGCTCTGGTCTTGGT) were obtained from Darmacon. These plasmids were puro resistance. The full-caPAK3 (T421E), a constitutively active mutation of PAK3, was cloned to pCDNA3.1-Hygro. H1299 cells were transfected with plasmid and selected at 400 ug/ml hygromycin B.

**Real-time quantitative RT-PCR.** Tissues and cells were homogenized in 1 ml RNAiso™ Plus lysis buffer (TAKARA). Total RNA was extracted and 2 μg RNA was transcribed into cDNA with M-MLV reverse transcriptase (Invitrogen) following the manufacturer's instruction. SYBR Green Premix Ex Taq (Takara) was used for quantitative RT-PCR analysis. The gene-specific primers are listed in Supplementary Table 1. Technique replicates were used for every sample and each experiment was performed at least two times.

**Immunohistochemistry.** The human sample study was approved by the independent ethics committee at the East China Normal University. All clinical samples were devoid of personal information. Lung cancer tumors, lung cancer metastasis tumors, normal lung samples, or mouse lung cancer metastasis tissues were fixed with 4% paraformaldehyde (Beijing Dingguo Changsheng Biotechnology co. LTD) for 12 h, and transferred into gradient ethanol, rolled, processed, and embedded into paraffin. Four-micrometer sections were cut on a microtome (Leica, Germany). Mounted sections onto charged slides and dried for 2 h at 62 °C. Blocked each sample with 100–400 μl blocking solution (NeoBioscience, ENS004.120; Boster, SA1053) for 30 min at room temperature to prevent nonspecific binding of the antibodies. Added primary antibodies in indicated dilutions to each sample and incubated overnight at 4° in a humidified chamber. Smad4 (Santa Cruz, sc-7966, clone: B-8, lot: H0212, 1:200), p-c-Jun (CST #2361, clone: 54B3, lot: 7, 1:100), p-JNK (CST #4668, clone: 81E11, lot: 11, 1:50), Ras (G12D) Mutant (CST #14429, clone: D8H7, lot: 1, 1:50), P53 (Santa Cruz sc-126, clone: DO-1, lot: L5479, 1:50), TTF1 (Abcam ab76013, clone: EP1584Y, lot: GR297063-7, 1:200). Covered sections with detection reagent for 20 min at RT. Then stained with DAB for 1–5 min at RT. To evaluate the staining intensity in IHC, the white-view pictures on the slides were

taken under the same condition. We utilized Image-Pro to select cells with the positive staining by calculating the gray values. The combination of size ratio between the gray areas and the whole field as well as staining intensity were calculated, and we utilized the resulting index as our definition of the staining intensity ($+$, $+++$, etc.). The negative ($-$) means the percentage of the positive cells with less than 25% and near background staining; the index of the positive cells between 25 and 50% with low intensity is considered as weak ($+$); those between 50 and 75% with intermediate staining refers to moderate staining ($++$); those higher than 75% or more than 50% with the strongest staining represents intense staining ($+++$).

**Western blotting analysis.** The cells or tissues were collected, washed with cold PBS buffer, and lysed using lysis buffer (RIPA buffer, 89900, Thermo Fisher) on ice for 30 min. Proteins were harvested from the lysates, and protein concentrations were quantified using BCA kit (Thermo Scientific #23227) following the instruction. Equal amount of protein from each sample was lysed in SDS sample buffer and resolved in 8–12% gradient SDS gels. Separated proteins were transferred to nitrocellulose membranes and immunoblotted with primary antibodies specific for PAK3 (CST #2609T, Clonality: Polyclonal, lot:2, 1:1000), c-Jun (CST #9165S, clone: 60A8, lot: 11, 1:1000), p-c-Jun (CST #2361, clone: 54B3, lot: 7, 1:1000), JNK (CST #9258S, clone: 56G8, lot: 11, 1:1000), p-JNK (CST #4668, clone: 81E11, lot: 11, 1:1000), Smad4 (CST #46535, clone: D3R4N, lot:2, 1:1000), GAPDH (Proteintech 60004-1-IG, clone: 1E6D9, lot: 10013030, 1:1000), p-MEK1 (Ser298) (Abcam, ab96379, clone: EPR3338, Lot: GR251630-1), MEK1 (Abcam, ab32576, clone: Y77, Lot: GR245617-1), E-Cadherin (CST, #3195, clone: 24E10, Lot:13), TWIST1 (CST, #69366, clone: E7E2G, Lot:1), SNAIL1 (Abcam, ab216347, clone: EPR21043, Lot: GR216730-1), β-actin (MBL M177-3, clone: 6D1, lot: 008, 1:5000) overnight at 4 °C. After incubation with a fluorescent-labeled secondary antibody (Invitrogen; Jackson Immunoresearch, 1:5000 dilutions), specific signals for proteins were visualized by a LI-COR Odyssey Infrared Imaging System. Alexa Fluor® 680 AffiniPure Goat Anti-Mouse IgG (H + L), Jackson ImmunoResearch, Code: 115-625-146, Clonality: Polyclonal, lot: 146048, RRID: AB_2338935. Alexa Fluor® 790 AffiniPure Goat Anti-Mouse IgG (H + L), Jackson ImmunoResearch, Code: 115-655-146, Clonality: Polyclonal, lot: 108636, RRID: AB_2338944. Alexa Fluor® 790 AffiniPure Goat Anti-Rabbit IgG (H + L), Jackson ImmunoResearch, Code: 111-655-144, Clonality: Polyclonal, lot: 134979, RRID: AB_2338086.

**Transwell assay.** Transwell assays were performed in 24-well PET inserts (Millipore, 8.0-μm pore size) for cell migration. $5 \times 10^4$ cells in serum-free media were

plated in the upper chamber of transwell inserts (Millipore PIRP12R48) (two replicas for each sample) for 12–20 h. The inserts were then placed into 10% serum media for indicated hours of migration as described in figures legend. Cells in the upper chambers were removed with a cotton swab, and migrated cells were fixed in 4% paraformaldehyde, and stained with 0.5% crystal violet. Filters were photographed, and the total migrated cells were counted. This experiment was repeated independently three times.

**Wound healing assay**. For wound healing assay, cells cultured in the 12-well plate were scratched by a small pipette tip to produce a "wound", and monitored the "healing" after 0, 6, or 12 h. The images of each well were captured, and the closure in each well were counted. Every experiment was repeated independently three times.

**F-actin staining**. The cover slides were treated with 0.01% Poly-L-Lysine (Sigma) for 30 min and 10 μg/mL Fibronectin (Corning) for 12 h, placed in a 24-well plate, and inoculated with $5 \times 10^4$ cells per well. The cells were switched to non-serum medium for 24 h, followed by 10% serum medium for 0, 15 min. The cells were washed with PBS at 37 °C, fixed by 4% paraformaldehyde, permeated by 0.25% Triton X-100, stained by Rhodamine-Phalloidin (Invitrogen) and DAPI (Thermo, avoid light). The coverslips were then mounted on slides for microscopic visualization.

**Luciferase assay**. pcDNA3.1-Luciferase-MCS vector was a gift from Dr. Ping Wang (Tongji University). pcDNA3.1-Luciferase reporter constructs were generated by cloning the 3'UTR of PAK3 (wild type/mutant) at the location of MCS of pcDNA3.1-Luciferase-MCS vector. Co-transfection of WT or mutant pcDNA3.1-Luciferase reporter constructs and miRNA mimics or negative control mimics into cells. After 48 h, removed medium and rinsed cells with PBS twice. Then used Luciferase Reporter Assay System (Promega) and record the firefly luciferase activity measurement.

**CHIP assay**. PK and SPK cells were transfected with flag-PRK5-SMAD4 or flag-PRK5 vector plasmids for 36 h, immersed in RPMI-1640 medium supplemented with 1% formaldehyde and protease inhibitor (Roche, USA) for 5 min. And then 125 mM glycine was added for 3 min. Cells were washed and harvested. Cells were resuspended in sonic buffer and then sonicated until clear to result in DNA fragments of 100–500 bp in length. Wash the protein G magnetic beads (Invitrogen, 10612D) with RIPA 0.3 buffer, add flag antibody (Abcom, ab49763), and rotated at 4° for 6 h. Then, 1 ml samples were mixed with Magnetic Protein G Beads, rotated at 4° overnight. Beads were washed with RIPA 0.3 buffer, RIPA 0 buffer, LiCl buffer, and TE buffer, respectively. Immunocomplexes were extracted from the beads with SDS elution buffer with RNase, and then added Proteinase K. Additionally, crosslinks were reversed at 65 °C for at least 6 h. Moreover, DNA fragments were purified with a DNA purification kit. Q-PCR was conducted to assess the enrichment fold of immunoprecipitated DNA during the ChIP experiments. The sequences of the primers used are provided in Supplementary Table 1.

**Bioinformatics analysis**. We downloaded the raw data from a published paper (PMID: 11707567) and then sorted the patients who had lung cancer primary site tumor ($n = 123$) and lung cancer metastasis tumor ($n = 7$). We analyzed the expression of SMAD4 and PAK3 of these patients' tumors and conducted the Oncomine analysis (https://www.oncomine.org/) on the published datasets by utilizing the cutoff criteria ($P < 0.05$; |log2 Fold change|>1.5).

**Statistical analysis**. Prism software (GraphPad Software) was used for statistical analyses. The intensity of the Western blot results was analyzed by densitometry using ImageJ software. Values were shown as mean ± s.e.m. Statistical significance between two samples was determined with two-tailed Student's $t$-tests.

**Study approval**. All mice studies were approved and used in accordance with institutional guidelines (East China Normal University). Human specimens used in this study have been approved by the East China Normal University.

**Reporting summary**. Further information on research design is available in the Nature Research Reporting Summary linked to this article.

## Data availability
The RNA-sequencing data have been deposited in the GEO database under the accession code GSE164436. The mRNA expression of SMAD4 and PAK3 of these patients' data referenced during the study are available in a public repository from the Oncomine website (https://www.oncomine.org/). The source data underlying Figs. 1–7 and Supplementary Figs. 1–10 is provided as a Source Data file. All the other data supporting the findings of this study are available within the article and its supplementary information files and from the corresponding author upon reasonable request. A reporting summary for this article is available as a Supplementary Information file.

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

## Acknowledgements

This work was supported in part by grants from the National Natural Science Foundation of China (31730017, 81672883, 81261120555, 31200878, 31071875, 81271742, and 31401012), Shanghai natural science foundation (19JC1411900, 17ZR1407900, and 14ZR1411400), and the Shanghai Sailing Program (21YF1431700).

## Author contributions

Conceptualization: X.L., J.L., X.Y., L.Li., and D.S.; Investigation: X.T., L.T., L.L., J.X., S.X., L.J., J.F., Q.L., S.S., Y.L., and Y.X.; Resources: D.S., X.Y., R.E.M., N.B., Y.W., J.Z., L.T., and K.-K.W.; Bioinformatics analyses: J.L. and F.G.; Funding acquisition and supervision: X.L., X.Y., J.L., and L.Li.; Revision: J.L., X.L., and L.Li.; Visualization and writing: J.L., X.L., and X.Y.

## Competing interests

The authors declare no competing interests.
