## [Peer Review File · Nature Communications]

Reviewers' comments:

Reviewer #1 (Expertise: Lung cancer models, metastasis, Remarks to the Author):

Authors established genetically engineered mouse model of lung cancer with KRASG12D, P53fl/fl, and/or smad4fl/fl LOF mutations. They demonstrated that in the presence of KRAS mutation and loss of P53, down-regulation of smad4 decreased expression of miR495 and miR543, inversely increased PAK3 (a target of miR495 and miR543) and thereby activated JNK-Jun signaling pathway which augmented metastatic potential. They also showed lower expression of smad4 and higher expression of PAK3, pJNK and pJun in metastatic lesions of lung cancer compared with normal tissues using their clinical specimens and TCGA data base.

Major comments

1. Majority of experiments were performed with tumors and tumor cell lines obtained from mice with KRASG12D, P53fl/fl, smad4fl/fl LOF mutations. However, lung cancer with triple mutations in KRAS, P53, and smad4, is unknown. The incidence of LOF smad4 mutations in lung cancer seems to be very rare. Although authors used reference 19 for smad4 mutation in NSCLC, the paper reported the smad4 mutation in colon cancer. Therefore, clinical relevance of this study may be very limited.
2. Smad4 is a major molecule in TGF- β /TGF β -R signaling which plays dual roles on tumor progression. TGF- β /TGF β -R signaling acts as a suppressor of tumorigenesis at early stages but promotes the tumorigenesis at the later stages, including metastatic stage (Morikawa M et al, *Oncogene* 2013, 32:1609). In the present study, authors demonstrated that loss of smad4 promoted metastasis in the presence of KRAS/p53 mutations. How authors explain the discrepancy of the role of TGF- β /TGF β -R signaling in metastatic process between the present study and previous studies?
3. Fig 7. Authors stated to examine 30 metastatic samples from lung cancer and 15 normal lung samples. However, information of these samples is lacking. In addition, number in Figure 7B seems to be incorrect and unreliable. For example, correct total number for SMAD4/lymph metastasis was 21, not 22. The number for SMAD4/bone metastasis was 9, not 8. Moreover, status of KRAS, P53, or SMAD4 should be shown. To show the role of SMAD4, PAK3, pJNK, and pJun, authors should compare primary tumor and metastatic tumors.
4. Fig 7, TCGA LUAD data. Bioinformatics analysis for SMAD4 and PAK3 expression comparing primary tumors and metastatic tumors may be helpful to increase the clinical relevance of results in the present study.

Minor comments.

1. Why H1299 cells were used? Characteristics of H1299 should be stated.

2. There are several typographical errors.

Reviewer #2 (Expertise: TGFb/SMAD signaling, cancer metastasis, Remarks to the Author):

This is an elegant study illustrating that Smad4 loss accelerates tumour initiation and metastatic progression of a mouse model of Kras mutation and p53 loss driven lung cancer. The authors dissect the mechanism of this and reveal that Smad4 upregulates miR495 and miR539 to suppress PAK3 expression which itself can act to activate JNK signalling to promote migration and invasion in-vitro. On the whole, the experiments are well performed and the authors should be congratulated on an excellent study. I have a few points that I would like them to address and consider.

1. It would be good to show an analysis of TCGA lung cancer data illustrating the frequency of Kras mutation, tp53 loss and Smad4 loss in human samples and the frequency of their co-occurrence and the lung cancer subtypes in which these events occur to illustrate what percentage and type of human lung cancer the authors are modelling.

2. With the above in mind, it would also be excellent if the authors could demonstrate which type/subtype of lung cancer the PK and SPK models represent.

3. The authors demonstrate that SPK mice have a higher primary tumor load and increased metastatic rate. They should also present their survival analysis shown in Fig1c as Kaplan-Meier survival curves with associated log rank statistics.

4. In Fig S1A the number and sites of metastasis are presented. It would be good to show this data in a table listing each mouse with metastasis so it can be assessed if some mice have lymph node only mets or some mice show mets to many sites for example.

5. In Figure S1d the authors show that SPK mice have a higher primary tumor burden. The authors should show if primary tumor burden correlates with metastatic burden. Do mice with more primary tumors in PK and SPK models have more mets?

6. The authors show IHC/RT-PCR for Smad4 in primary tumors of PK and SPK mice. Do Smad4 levels reduce in the mets in PK mice?
6. The authors should add biological n numbers to all figures where graphs have been presented. This is missing from most of these analyses.
7. All figures with images presented should have a scale bar added and referred to in the legend. This is provided for some but not all of these figures.
8. In Figure S2 the analysis is presented as if PK1 and SPK1 cell lines are matched. This is not the case so all crosswise comparisons should be analysed.
9. The effect of Smad4 knockdown on H1299 cells on cell migration and invasion should be measured.
10. RNAseq data shown in Fig 3A should be provided as a supplementary table.
11. In Fig S3D is it only 1-4% of H1299 cells invade in the transwell assay?
12. It would be good to have a figure showing the homology of the 3'UTR regions of murine and human PAK3
13. In Figure 7 the authors importantly analyse human correlations of their findings. In Samples analysed in Fig 7B the authors should analyse the correlations of Smad4 expression, PAK3 expression and p-JUN in a lesion-by-lesion basis to reveal if Smad4 low tumors statistically have higher PAK3 and higher p-JNK and p-Jun levels. Similarly, in Figure 7C the authors should also analyse the correlation of PAK3 and SMAD4 expression in a sample-by-sample basis.

Reviewer #3 (Expertise: PAK signaling, cancer, Remarks to the Author):

This work examines the signaling pathways that link loss of Smad4 to gain of Pak3/Jnk/motility in lung cancer mouse models and cells. According to the model, loss of Smad4 in a Kras/TP53 GEMM leads to loss of expression of various miRs including a few that regulate Pak3 expression. The data on this point are reasonably clear. The authors then show a correlation of Pak3 expression with Jnk activation and cell motility and metastasis. If all true, then the authors may have uncovered a new potential target for therapy in certain lung cancers.

The work is incomplete as presented, however, as detailed below:

1) The focus on Pak3 is poorly justified. It seems that the expression of many genes was altered by loss of Smad4 - why the sudden jump to Pak3? There is very little discussion of the data. The expression profile should be available in the Supplemental section and some justification given about why Pak3, among all the mRNAs, was chosen to study. Pathway analyses should also be shown.

2) What about the other Paks? Was their expression also altered? Pak2 has been implicated previously in TGFb signaling, and Pak1 in cell motility.

3) There is no demonstration of Pak activity in any of the data.

4) There is scant information about the ca Pak3 mutant. What is the mutation and how do we know its constitutively active. Such clones are notoriously difficult to maintain in bacteria.

5) Does exogenous expression of Pak3 override loss of Smad2? I ask this because so many miRs are regulated by this factor and, while correlations are shown, causality is not proven by their experiments.

6) the in vivo data from the mouse model i Figure 6 are arguably the most important findings in the paper, but I'm not willing to bank too much on a difference between 0/5 versus 2/5 metastases.

2

Reviewer #4 (Expertise: Lung cancer genomics, Remarks to the Author):

The authors of this study investigate the effect of homozygous loss of Smad4 in a Kras/Trp53 driven lung cancer model (KrasG12D, Trp53flox/flox). The authors present data to argue that loss of Smad4 in this context induces microRNA expression and thereby indirectly controls the level of PAK3 expression. Increased expression of PAK3 in turn, leads to increased JNK and Jun phosphorylation

and the authors conclude that this signalling cascade is the cause for the increased metastatic frequency observed.

In general, the manuscript is clearly written and easy to read, unfortunately the methods section is underdeveloped with vital information missing as well as multiple methods missing completely, which would make this study very difficult to reproduce.

Although the findings presented here are potentially interesting, I am not convinced that the data presented supports the conclusions. There are several key issues that need to be addressed before the manuscript might be suitable for publication.

First, and most importantly, the authors have not provided clear evidence that miR-495 and miR-543 are direct targets of Smad4 activation, by for instance ChIP, indeed, there is also no data showing whether Smad4 is a direct regulator of PAK3. The authors say on page 7 that “we could not detect any direct association between Smad4 and the Pak3 gene (data not shown). As no mention is made by which method this was investigated the statement is difficult to evaluate. In addition, Smad4 binding within the PAK3 gene has been demonstrated by ChIP-seq in ovarian cancer (Kennedy et al, PLOS1 2011).

Second, throughout the manuscript the authors use wound-healing assays to investigate the effects of Smad4 loss. I am very surprised by the varying quality of the data. In figure 2c, 6g and S6g the data is very convincing with good quality images. It is unfortunate that the same can not be said about figures 3f, 6f and S6f. It is also very unclear how the cells used in figure 2 were isolated and made immortal as no mention of these procedures are to be found in the manuscript.

Third, given the importance of the RNA-seq data and the conclusions thereof (upregulation of metastasis associated genes) it would be nice to be able to read the names of said genes. This might be an effect of making the PDF, but the authors should be aware of this potential issue (Figure 3a). Also in figure 3b, it is not clearly stated how the 9 samples are derived, are these 9 tumours from one mouse or from 9 different mice? It would also be nice to have the blot showing the differences in PAK3 protein levels in this figure.

Fourth, it is somewhat surprising that the authors conclusion that the effect of PAK3 activation is an upregulation of Jun phosphorylation as figure 4b shows a clear upregulation in total Jun protein in SPK cells. The same upregulation can be seen in figure S4 after PAK3 overexpression and conversely a reduction of cJun(?) protein levels is observed upon short hairpin inactivation of PAK3. I assume the authors mean cJun, there is also no mention in the manuscript if JAK1,2 or 3 is being

investigated. The methods section is no help as the catalogue numbers for the specific anti bodies are not listed and indeed, the total Jun/Jnk antibodies used are not mentioned at all. Again all information with regards to the cell line generation as well the hairpins used is missing. Figure 4b is also missing the Smad4 expression control. This is also true for figure 5c-e. As a minor point the scale in figure 4c is missing.

Fifth, in figure S5B the authors show a dose dependant increase of miR expression after stimulation. The data would be much clearer if it was stated what was used to stimulate the cells as Smad4 is an intracellular mediator of not only TGFbeta- but also for BMP family members, and indeed Smad4 loss has been associated with dysregulated BMP signalling and metastasis in colorectal cancer (Vorneveld et al 2014).

Additional comments:

The correct nomenclature for the mouse p53 gene is Trp53.

The white balance in figure S1 makes it impossible to see the tumours. Better photos might perhaps be chose.

Reviewers:

Mariam Jamal-Hanjani and Eva Gronroos

Response letter

Dear Reviewers,

We have tried our best to complete a series of experiments suggested by the Reviewers. Altogether 68 pieces of new or revised data have been incorporated into the revised manuscript. We hope that this revised manuscript can be qualified for publication at *Nature Communications*. Below are our point-by-point responses to the reviewers' comments.

Arrangement of Figures

Original	In Revision
Fig. 1A	Fig. 1A (new)
Fig. 1E	Fig. 1E (new)
Fig. 3A	Fig. 3A (new)
Fig. 3B	Fig. 3B (new)
	Fig. 3C (new)
Fig. 3C	Fig. 3D (relocation)
Fig. 3D	Fig. 3E (new)
Fig. 3E	Fig. 3F (relocation)
Fig. 3F	Fig. 3G(new)
Fig. 3G	Fig. 3H (new)
Fig. 3I	Fig. 3I (new)
Fig. 4A	Fig. 4A (new)
Fig. 4B	Fig. 4B (new)
Fig. 4D	Fig. 4D(new)
Fig. 5C	Fig. 5C (new)
Fig. 5D	Deleted
Fig. 5E	Fig. 5D (new)
Fig. 5F	Fig. 5E (new)
Fig. 5G	Fig. 5F (new)
	Fig. 5G (new)
	Fig. 5H (new)
	Fig. 5I (new)
Fig. 6D	Fig. 6D (new)
Fig. 6E	Fig. 6E(new)
Fig. 6F	Fig. 6F (new)
Fig. 6J	Fig. 6J (new)
Fig. 7A	Fig. 7A (new)
Fig. 7B	Fig. 7B (new)

Fig. 7C
Fig. 7D

Fig. 7C (new)
Fig. 7D (new)

Supplement Figures

Original

Fig. S1A
Fig. S1C
Fig. S1D

Fig. S2A
Fig. S2B

Fig. S3C
Fig. S3D
Fig. S3E
Fig. S3F

Fig. S5A
Fig. S5B
Fig. S5C
Fig. S5D
Fig. S5E
Fig. S5F

Fig. S6A
Fig. S6B
Fig. S6C
Fig. S6D
Fig. S6E
Fig. S6F
Fig. S6G
Fig. S6I

In Revision

Fig. S1A (new)
Fig. S1C (new)
Fig. S1D (new)
Fig. S1F (new)
Fig. S1G (new)
Fig. S3E (new)
Fig. S2A (new)
Fig. S2B (new)
Fig. S2F (new)
Fig. S3C (new)
Fig. S3D (new)
Fig. S3E (new)
Fig. S3F (new)
Fig. S3H (new)
Fig. S4D (new)
Fig. S4E (new)
Fig. S4F (new)
Fig. S4G (new)

Fig. S5A (new)
Fig. S5B (new)
Fig. S5C (new)
Fig. S5D (relocation)
Fig. S5E (new)
Fig. S5F (new)
Fig. S5G (new)
Fig. S6A (relocation)
Fig. S6B (relocation)
Fig. S6C (relocation)
Fig. S6D (relocation)
Fig. S6E (relocation)
Fig. S6F (relocation)
Fig. S6G (new)
Fig. S6H (new)
Fig. S7A (relocation)
Fig. S7B (new)

Fig. S7C (relocation)
Fig. S7D (relocation)
Fig. S7E (new)
Fig. S7F (new)
Fig. S7G (new)
Fig. S7I (new)
Fig. S8A (new)
Fig. S8B (new)
Fig. S9A (new)
Fig. S9B (new)
Fig. S9C (new)
Fig. S9D (new)

Reviewers' comments:

Reviewer #1 (Expertise: Lung cancer models, metastasis, Remarks to the Author):

Authors established genetically engineered mouse model of lung cancer with KRASG12D, P53fl/fl, and/or smad4fl/fl LOF mutations. They demonstrated that in the presence of KRAS mutation and loss of P53, down-regulation of smad4 decreased expression of miR495 and miR543, inversely increased PAK3 (a target of miR495 and miR543) and thereby activated JNK-Jun signaling pathway which augmented metastatic potential. They also showed lower expression of smad4 and higher expression of PAK3, pJNK and pJun in metastatic lesions of lung cancer compared with normal tissues using their clinical specimens and TCGA data base.

Major comments

1. Majority of experiments were performed with tumors and tumor cell lines obtained from mice with KRASG12D, P53fl/fl, smad4fl/fl LOF mutations. However, lung cancer with triple mutations in KRAS, P53, and smad4, is unknown. The incidence of LOF smad4 mutations in lung cancer seems to be very rare. Although authors used reference 19 for smad4 mutation in NSCLC, the paper reported the smad4 mutation in colon cancer. Therefore, clinical relevance of this study may be very limited.

Response: Thank you for your comments!

Here we conducted more bioinformatics analyses of the SMAD4 mutation. 3.4% SMAD4 mutations (n = 57) can be found in human lung tumors (n = 1668) by analyzing the MSK-IMPACT datasets (PMID: 28481359; *Nature Medicine*. 2017). We have included detailed information in the Sup. Table 2. Using the same group of patient tumors, we found that 8 of 57 patients have triple mutations in KRAS, TP53, and SMAD4. We included this detail information in the Sup. Table 3. Considering that lung cancer is the leading cancer killer worldwide (~2

million deaths per year), we think our findings from the current mouse models are relevant to many patients.

We also add a related paragraph in the Discussion.

“Considering that some human lung tumors have not only SMAD4 mutation but also triple mutations in KRAS, TP53, and SMAD4 (Sup. Tables 2-3)⁵⁴, our findings may address the impact of p53 mutation on Smad4 LOF in the progression of these lung cancers.”

2. Smad4 is a major molecule in TGF- β /TGF β -R signaling which plays dual roles on tumor progression. TGF- β /TGF β -R signaling acts as a suppressor of tumorigenesis at early stages but promotes the tumorigenesis at the later stages, including metastatic stage (Morikawa M et al, Oncogene 2013, 32:1609). In the present study, authors demonstrated that loss of smad4 promoted metastasis in the presence of KRAS/p53 mutations. How authors explain the discrepancy of the role of TGF- β /TGF β -R signaling in metastatic process between the present study and previous studies?

Response: We believe that SMAD4-dependent regulation of PAK3 is TGF- β independent. We demonstrate (in Figure S3A) PAK3 levels were not affected by depletion of Smad3 or by a time-course treatment of TGF- β in H1299 cells. Therefore, effect on miRNA-PAK3 pathway might be a moonlight function of Smad4, which deserves further investigation for its association with early and late stage of cancer development.

Morikawa M *et al.* mentioned (Oncogene 2013, 32:1609) that the dual role of transforming growth factor β (TGF- β) is dependent on the specific cellular context. The presence of *Kras/Trp53* mutations, a *de novo* research background for the role of SMAD4 in lung tumor development and metastasis, may reflect a unique cellular context and partially reconcile the discrepancy of the role of TGF- β /TGF β -R signaling in the metastatic process between the present study and previous ones.

3. Fig 7. Authors stated to examine 30 metastatic samples from lung cancer and 15 normal lung samples. However, information of these samples is lacking. In addition, number in Figure 7B seems to be incorrect and unreliable. For example, correct total number for SMAD4/lymph metastasis was 21, not 22. The number for SMAD4/bone metastasis was 9, not 8. Moreover, status of KRAS, P53, or SMAD4 should be shown. To show the role of SMAD4, PAK3, pJNK, and pJun, authors should compare primary tumor and metastatic tumors.

Response: According to the reviewer's comments, we listed the point-by-point responses below and the related figures.

- 1) Here we included the detailed information of these patients, and please find the information in the Sup. Table 4.
- 2) Sorry for the mistake. We had 22 patients with SMAD4/lymph metastasis and 8 patients for SMAD4/bone metastasis. More detailed information can be found in the Sup. Table 4.
- 3) We have shown expression of SMAD4, PAK3, pJNK, and p-c-Jun in the Figures 7A-B

and included expression of RAS (G12D) and p53 in the Figures S9A-B. More detailed information can be found in the Sup. Table 4.

- 4) We included two new groups of patients, lung tumors with or without metastases, to examine the expression profiles of SMAD4, PAK3, pJNK, and p-c-Jun, in addition to three original groups (Normal, Lymph metastasis, and bone metastasis) in the revised Figures 7A-B. More detailed information can be found in the Sup. Table 4.

We have revised the manuscript accordingly.

4. Fig 7, TCGA LUAD data. Bioinformatics analysis for SMAD4 and PAK3 expression comparing primary tumors and metastatic tumors may be helpful to increase the clinical relevance of results in the present study.

Response: Thank you for your suggestion!

We explored the possible clinical significance of SMAD4, PAK3, p-JNK, and p-Jun expression in human lung cancers, using samples of 15 normal controls, 15 early tumors, 12 advanced tumors, and 30 metastatic human lung cancers (Sup. Table 4). We found that SMAD4 was highly expressed in the controls, indicated by (+++), while its expression decreases during the progression from early tumors to metastatic ones. In contrast, the levels of PAK3, p-JNK, and p-JUN are higher in primary tumors and metastatic ones, compared with controls (Fig. 7A and 7B).

Moreover, bioinformatics analysis of a published human dataset³⁵ demonstrate an overall reduction in SMAD4 expression and an elevated PAK3 expression in 7 metastatic lung tumor tissues compared with 123 primary lung tumor samples (Fig. S9C).

Minor comments.

1. Why H1299 cells were used? Characteristics of H1299 should be stated.

Response: H1299 is a cell line lack of p53, containing RAS mutation. It is compatible with our mouse model for in vitro analysis of Smad4 functions. Information is added in page 6.

2. There are several typographical errors.

Response: We have corrected.

Reviewer #2 (Expertise: TGFb/SMAD signaling, cancer metastasis, Remarks to the Author):

This is an elegant study illustrating that Smad4 loss accelerates tumour initiation and metastatic progression of a mouse model of Kras mutation and p53 loss driven lung cancer. The authors dissect the mechanism of this and reveal that Smad4 upregulates

miR495 and miR539 to suppress PAK3 expression which itself can act to activate JNK signalling to promote migration and invasion in-vitro. On the whole, the experiments are well performed and the authors should be congratulated on an excellent study. I have a few points that I would like them to address and consider.

1. It would be good to show an analysis of TCGA lung cancer data illustrating the frequency of Kras mutation, tp53 loss and Smad4 loss in human samples and the frequency of their co-occurrence and the lung cancer subtypes in which these events occur to illustrate what percentage and type of human lung cancer the authors are modelling.

Response: Thank you for your comments!

Here we have conducted new bioinformatics analyses of SMAD4 mutations in lung cancers. 3.4% SMAD4 mutations (n = 57) can be found in human lung tumors (n = 1668) by analyzing the MSK-IMPACT datasets (PMID: 28481359; *Nature Medicine*. 2017). We have included detailed information in the Sup. Table 2. In the same group of patient samples, we have found that 8 of 57 have triple mutations in *KRAS*, *TP53*, and *SMAD4*. The detailed information has been included in Sup. Table 3. Given that lung cancer is the leading cause of cancer related death worldwide (~2 million deaths per year), we believe our findings from the current mouse models are relevant to significant amount of lung cancer patients. We also add a new paragraph in the Discussion.

We also add a related paragraph in the Discussion.

“Considering that some human lung tumors have not only SMAD4 mutation but also triple mutations in KRAS, TP53, and SMAD4 (Sup. Tables 2-3)⁵⁴, our findings may address the impact of p53 mutation on Smad4 LOF in the progression of these lung cancers.”

2. With the above in mind, it would also be excellent if the authors could demonstrate which type/subtype of lung cancer the PK and SPK models represent.

Response: This is a great comment. Considering the general observations of our mouse tumors, including pathological morphology, marker staining, and RNA-Seq analysis, our PK and SPK mouse models are likely to represent human lung adenocarcinoma. In detail, the cells in lung tumors of both mouse models resemble gland cells, a typical morphology of adenocarcinoma. It is further supported by a positive staining of TTF1, a marker of lung adenocarcinoma (Listed below). Also, our RNA-Seq analyses of PK and SPK tumors demonstrate a similar expression pattern of TTF1 among all the tumor cells (Table 1). Meanwhile, we observed few sporadic lung tumors (~20%) resembling a feature of squamous cell carcinoma, such as the typical nests of neoplastic squamous cells with positive staining p63 and Krt5, shown in Figure S1G. We employed “lung cancer” as the general term for our mouse lung tumors in case mixed population of additional subtypes may exist.

Part of Table 1 (Lung Adenocarcinoma marker: TTF1; Lung Squamous Cell Carcinoma marker: Krt5, SOX2, and Trp63)

gene_id	Expression_pk1	Expression_pk2	Expression_pk3	Expression_spk1	Expression_spk2	Expression_spk3
Ttf1	3.38809	3.40657	3.09721	3.68543	3.54401	3.77513
Krt5	0.306589	0.43149	0.684918	0	0	0
Sox2	0	0	0	0	0.427328	0.448323
Trp63	0	0	0	0.0134092	0.0195926	0.0138678

3. The authors demonstrate that SPK mice have a higher primary tumor load and increased metastatic rate. They should also present their survival analysis shown in Fig1c as Kaplan-Meier survival curves with associated log rank statistics.

Response: According to the reviewer's suggestion, we have included the Kaplan-Meier survival curves, showing a significant difference in survival rate between PK and SPK mice. Please refer to Figure S1F.

4. In Fig S1A the number and sites of metastasis are presented. It would be good to show this data in a table listing each mouse with metastasis so it can be assessed if some mice have lymph node only mets or some mice show mets to many sites for example.

Response: Thanks, we have listed all PK or SPK mice with single or multiple site metastasis in the new Sup. Table 1 (lung cancer metastasis of PK &SPK mouse).

5. In Figure S1d the authors show that SPK mice have a higher primary tumor burden. The authors should show if primary tumor burden correlates with metastatic burden. Do mice with more primary tumors in PK and SPK models have more mets?

Response: We didn't observe that higher primary tumor burden in both PK and SPK models correlates with more metastases, compared with those with lower primary tumor burden.

6. The authors show IHC/RT-PCR for Smad4 in primary tumors of PK and SPK mice. Do Smad4 levels reduce in the mets in PK mice?

Response: We did the immunohistochemical staining of SMAD4 protein in metastatic tissues of PK and SPK mice. We found that the expression of Smad4 was barely detectable in metastasized lesion in PK or SPK mice (Fig 7A).

6. The authors should add biological n numbers to all figures where graphs have been presented. This is missing form most of these analyses.

Response: We've added the biological n numbers to all figures where graphs have been presented.

7. All figures with images presented should have a scale bar added and referred to in

the legend. This is provided for some but not all of these figures.

Response: We've added the scale bar in all figures with images.

8. In Figure S2 the analysis is presented as if PK1 and SPK1 cell lines are matched. This is not the case so all crosswise comparisons should be analysed.

Response: You are right, the PK1 and SPK1 cell lines are not matched. Therefore, we have included an additional pair (PK2 and SPK2) of cell lines (Fig. S2 A and S2B). Two pairs of cell lines displayed similar results.

9. The effect of Smad4 knockdown on H1299 cells on cell migration and invasion should be measured.

Response: We appreciate your suggestion. The effect of Smad4 knockdown on H1299 cells on invasion was measured previously. Knockdown of Smad4 on H1299 cells promoted cell invasion (Fig. S2F). The results are consistent with the negative impact of Smad4 overexpression on cell invasion (Fig 2E-F).

10. RNAseq data shown in Fig 3A should be provided as a supplementary table.

Response: We have included RNA-Seq data in Figure 3A, and Table 1, including a complete list of differentially expressed genes between SPK and PK cells.

11. In Fig S3D is it only 1-4% of H1299 cells invade in the transwell assay?

Response: Sorry for previous mislabel/misrepresentation (intended to show relative proportion of invasion). In the revised version, we have included more representative data (Fig. S3C-D).

12. It would be good to have a figure showing the homology of the 3'UTR regions of murine and human PAK3

Response: We have included the figure showing that the homology of the 3'UTR regions of murine and human PAK3 was 91% (Fig. S5G).

13. In Figure 7 the authors importantly analyse human correlations of their findings. In Samples analysed in Fig 7B the authors should analyse the correlations of Smad4 expression, PAK3 expression and p-JUN in a lesion-by-lesion basis to reveal if Smad4 low tumors statistically have higher PAK3 and higher p-JNK and p-Jun levels. Similarly, in Figure 7C the authors should also analyse the correlation of PAK3 and SMAD4

expression in a sample-by-sample basis.

Response: According to the reviewer's suggestions, we have performed Pearson's correlation analysis between SMAD4 and PAK3 or P-JNK or P-Jun expression in Figure 7B. Their R values are -0.29, -0.77, and -0.73, respectively (Fig. 7C). And there are highly positive correlation R- values between PAK3 and P-JNK or P-Jun (Fig. 7C). We add these results in Figure 7B. We also did the Pearson's correlation analysis between SMAD4 and PAK3 for Figure 7C. It shows a negative correlation between them ($R = -0.7652$, $P < 0.05$) (Fig. S9D).

Reviewer #3 (Expertise: PAK signaling, cancer, Remarks to the Author):

This work examines the signaling pathways that link loss of Smad4 to gain of Pak3/Jnk/motility in lung cancer mouse models and cells. According to the model, loss of Smad4 in a Kras/TP53 GEMM leads to loss of expression of various miRNAs including a few that regulate Pak3 expression. The data on this point are reasonably clear. The authors then show a correlation of Pak3 expression with Jnk activation and cell motility and metastasis. If all true, then the authors may have uncovered a new potential target for therapy in certain lung cancers.

The work is incomplete as presented, however, as detailed below:

1) The focus on Pak3 is poorly justified. It seems that the expression of many genes was altered by loss of Smad4 - why the sudden jump to Pak3? There is very little discussion of the data. The expression profile should be available in the Supplemental section and some justification given about why Pak3, among all the mRNAs, was chosen to study. Pathway analyses should also be shown.

Response: Following the reviewer's suggestion, we have reorganized the data with additional analysis (Figures 3A-C and Tables 1-4) that rationalizes our selection of PAK3. Specifically, we did KEGG pathway analyses of 3,777 differentially expressed genes (DEGs) between SPK and PK cells (Figure 3A and Table 1). And then, we found that these DEGs were highly enriched in cancer development (Table 2) and cell motility (Table 3) (Figure 3B). PAK3 is among the top differentially expressed genes overlapped between cancer development and cell motility datasets (Figure 3C and Table 4). Related discussion has been included in the manuscript.

2) What about the other Paks? Was their expression also altered? Pak2 has been implicated previously in TGF β signaling, and Pak1 in cell motility.

Response: We measured the expression of PAKs (PAK1, PAK2, and PAK3) in PK and SPK cells with or without the treatment of TGF β or TGF β in the presence or absence of TGF β inhibitor SB compound. The results shown below (Figure S3H) demonstrate that only the PAK3, but not PAK1 or PAK2, mRNA levels are significantly upregulated upon loss of SMAD4 in a TGF β -independent manner. The TRI gene, a target of TGF β signaling, is included here as

a positive control (Figure S3H, below). Additionally, we transfected PK cells with antagomir miR495/miR543 and found that only PAK3 mRNA was up-regulated (Figure S5F).

3) There is no demonstration of Pak activity in any of the data.

Response: We highly appreciate your suggestion. Activation of JNK-Jun signaling by PAK3 (same as other PAK proteins) has been well documented to reflect PAK activity (Liu et al., 2010; Endocr., Fanger, et al., 1997, EMBO, Rousseau, et al., 2002, JBC). We provided evidence for PAK3-mediated regulation of JNK-Jun pathway in loss-of-function and gain-of-function experiments (Sup. Figures S4B-C).

4) There is scant information about the ca Pak3 mutant. What is the mutation and how do we know its constitutively active. Such clones are notoriously difficult to maintain in bacteria.

Response: We highly appreciate your suggestion. The constitutively active PAK3 mutation (T421E) has been reported in numerous literatures (Rousseau et al., 2003, JBC. Manser et al., 1997, Mol. Cell. Biol., Sells, et al. 1997, Curr. Biol.). In the current study, we are lucky to express this clone in bacteria and mammalian cells with expected results (Fig. S3D-G; Fig S4C).

We also added the information about the ca-PAK3 mutant in the Method section and listed it below.

“These plasmids were puro resistance. The full-caPAK3 (T421E), a constitutively active mutation of PAK3, was cloned to pCDNA3.1-Hygro. H1299 cells were transfected with plasmid and selected at 400ug/ml hygromycin B.”

5) Does exogenous expression of Pak3 override loss of Smad2? I ask this because so many miRs are regulated by this factor and, while correlations are shown, causality is not proven by their experiments.

Response: We believe that SMAD4-dependent regulation of PAK3 is TGF- β independent. We demonstrated (in Figure S3A) that PAK3 levels were not affected by depletion of Smad3 or by a time-course treatment of TGF- β in H1299 cells. Moreover, our PAK3 promoter-luciferase showed that exogenous expression of SMAD4 or SMAD3 or SMAD2 did not affect mouse PAK3 promoter activity (Figure S4F). Therefore, we focused on investigating if exogenous expression of PAK3 overrides loss of Smad4, not Smad2 or Smad3. Here, we found that PAK3 knockdown indeed overrides the effect of “loss of Smad4” on cell mobility in SPK cells (Fig. 3F-H, S3, S4B-C).

6) the in vivo data from the mouse model i Figure 6 are arguably the most important findings in the paper, but I'm not willing to bank too much on a difference between 0/5

versus 2/5 metastases.

Response: This is a critical question! The original design was due to concern about expensive synthetic agomir. We have expanded our experiments to include additional groups of animals. Now we have observed a more significant difference between 0/10 versus 4/9 metastases (Fig. 6J). Fisher's exact test is applied to analyze the differences between the two groups, showing a statistical significance (two-tailed P value equals 0.0325).

Reviewer #4 (Expertise: Lung cancer genomics, Remarks to the Author):

The authors of this study investigate the effect of homozygous loss of Smad4 in a Kras/Trp53 driven lung cancer model (KrasG12D, Trp53flox/flox). The authors present data to argue that loss of Smad4 in this context induces microRNA expression and thereby indirectly controls the level of PAK3 expression. Increased expression of PAK3 in turn, leads to increased JNK and Jun phosphorylation and the authors conclude that this signalling cascade is the cause for the increased metastatic frequency observed.

In general, the manuscript is clearly written and easy to read, unfortunately the methods section is underdeveloped with vital information missing as well as multiple methods missing completely, which would make this study very difficult to reproduce.

Response: We highly appreciate your suggestion. We have revised the method section with detailed information.

Although the findings presented here are potentially interesting, I am not convinced that the data presented supports the conclusions. There are several key issues that need to be addressed before the manuscript might be suitable for publication.

First, and most importantly, the authors have not provided clear evidence that miR-495 and miR-543 are direct targets of Smad4 activation, by for instance ChIP, indeed, there is also no data showing whether Smad4 is a direct regulator of PAK3. The authors say on page 7 that "we could not detect any direct association between Smad4 and the Pak3 gene (data not shown). As no mention is made by which method this was investigated the statement is difficult to evaluate. In addition, Smad4 binding within the PAK3 gene has been demonstrated by ChIP-seq in ovarian cancer (Kennedy et al, PLOS1 2011).

Response: Thanks for your suggestion. We performed suggested experiments and SMAD4 ChIP-qPCR analyses show that SMAD4 can be recruited to genes encoding miR-495 and miR-543 in SPK cells (Fig. 5G-H). In addition, we have also shown that SMAD4 negatively regulated the expression of miR-495 and miR-543 (Fig. 5A-F, and S6A-F). These data demonstrate that miR-495 and miR-543 are direct targets of Smad4.

Given that ChIP-seq has demonstrated SMAD4 binding within the PAK3 gene in ovarian cancer (Kennedy et al., PLOS1 2011), we analyzed the SMAD4 ChIP-Seq in human lung cells (GSM1246719) or mouse lungs (GSM1376735). Unfortunately, we did not observe SMAD4 binding on the PAK3 promoter region in lung tumors (Figure S4D-E), perhaps due to tissue specificity. Moreover, our PAK3 promoter (2kb) -luciferase showed that exogenous expression of SMAD4 or SMAD3 or SMAD2 did not affect mouse PAK3 promoter activity (Figure S4F), suggesting that there is no direct regulation of PAK3 by SMAD4 via promoter regions.

Second, throughout the manuscript the authors use wound-healing assays to investigate the effects of Smad4 loss. I am very surprised by the varying quality of the data. In figure 2c, 6g and S6g the data is very convincing with good quality images. It is unfortunate that the same cannot be said about figures 3f, 6f and S6f. It is also very unclear how the cells used in figure 2 were isolated and made immortal as no mention of these procedures are to be found in the manuscript.

Response: We have replaced the ambiguous figures with more representative ones after several more repeating experiments (new Fig. 3G-H; Fig. 6F-G; and Fig. S7E-F). Procedures for generation of immortalized cells are included in revised manuscript.

Third, given the importance of the RNA-seq data and the conclusions thereof (upregulation of metastasis associated genes) it would be nice to be able to read the names of said genes. This might be an effect of making the PDF, but the authors should be aware of this potential issue (Figure 3a). Also in figure 3b, it is not clearly stated how the 9 samples are derived, are these 9 tumours from one mouse or from 9 different mice? It would also be nice to have the blot showing the differences in PAK3 protein levels in this figure.

Response: We have updated the original Fig. 3a-b with new Fig. 3A-C, listing the top differentially expressed genes. We examined the PAK3 protein and RNA expression in SPK lung tumors compared to PK tumors by immunohistochemical staining and quantitative PCR (Fig. 3D, 3E). To avoid confusion, we have removed the sub-figure with 9 samples.

Fourth, it is somewhat surprising that the authors conclusion that the effect of PAK3 activation is an upregulation of Jun phosphorylation as figure 4b shows a clear upregulation in total Jun protein in SPK cells. The same upregulation can be seen in figure S4 after PAK3 overexpression and conversely a reduction of cJun(?) protein levels is observed upon short hairpin inactivation of PAK3. I assume the authors mean cJun, there is also no mention in the manuscript if JAK1,2 or 3 is being investigated. The methods section is no help as the catalogue numbers for the specific anti bodies are not listed and indeed, the total Jun/Jnk antibodies used are not mentioned at all. Again all information with regards to the cell line generation as well the hairpins used is missing. Figure 4b is also missing the Smad4 expression control. This is also true for

figure 5c-e. As a minor point the scale in figure 4c is missing.

Response: Thanks for your suggestions. The effect of PAK3 activation is an upregulation of JNK phosphorylation, and downstream effector P-c-Jun. Activation of JNK-Jun signaling by PAK3 (same as other PAK proteins) has been well documented (Liu et al., 2010; Endocr., Fanger, et al., 1997, EMBO, Rousseau, et al., 2002, JBC).

We found the c-Jun mRNA level was up-regulated in SPK cells compared to PK cells (Fig. S4G). These results suggest that c-Jun expression can be a readout for SMAD4 depletion. In addition, we also observed elevated p-c-Jun in SPK cells, suggesting c-Jun protein is activated. These results support the conclusion that PAK3 overexpression activates the JNK-Jun signaling. Here we also made revisions accordingly in the manuscript.

We examined whether other PAK genes were regulated upon loss of SMAD4 expression. We found that SMAD4 depletion primarily affected expression of PAK3, but not PAK1 and PAK2 in mouse-derived cells (Fig. S3H).

We highly appreciate your suggestion. We have already added information about study materials in Methods, including antibody purchase and cell line generation. Meanwhile, we've repeated these Fig. 4A-B and Fig. 5C-D along with SMAD4 expression controls (Fig S5A-C). In addition, The scale bar of all pictures has already been added.

Fifth, in figure S5B the authors show a dose dependant increase of miR expression after stimulation. The data would be much clearer if it was stated what was used to stimulate the cells as Smad4 is an intracellular mediator of not only TGFbeta- but also for BMP family members, and indeed Smad4 loss has been associated with dysregulated BMP signalling and metastasis in colorectal cancer (Vorneveld et al 2014).

Response: Sorry for the confusion. We used TGF β to treat cells in Figure S5B and made the changes accordingly in the text.

“Screening indicated that three miRNAs, miR-495, miR-539, and miR-543, were positively regulated by SMAD4 in a dose-dependent manner in H1299 cells under TGF β treatment (Fig. S5E).”

Additional comments:

The correct nomenclature for the mouse p53 gene is Trp53.

The white balance in figure S1 makes it impossible to see the tumours. Better photos might perhaps be chose.

Response: Thank you for your suggestion! We changed the pictures of mouse lung tumors and corrected the nomenclature of the mouse p53 gene into Trp53. We have modified the

contrast for the figures in Fig. S1.

REVIEWER COMMENTS

Reviewer #1 (Remarks to the Author):

Authors examined the role of Smad4 (loss of function) in genetically engineered mouse models for lung cancer. They found that Smad4 loss of function with KRAS and TP53 (loss of function) mutations facilitated the production of metastasis. The metastasis was associated with activation of PAC3, which stimulated tumor cell motility, due to downregulation of miR-495 and miR-543. They further showed inverse correlation between Smad4 and PAK3/pJNK/pJun in clinical specimens obtained from lung cancer patients. The mechanism by which Smad4 loss of function resulted in increase of metastasis was clearly shown. However, there are several issues need to be addressed.

Major comments

1. How authors evaluated the production of metastasis in mouse models? The methods for detection of metastasis was not stated in the manuscript.
2. How authors evaluated the staining intensity in IHC. The definition of the staining intensity (+, +++, etc) was not stated in the manuscript.
3. Figure 7. 15 early, 12 advanced, and 30 metastatic lung cancer specimens were evaluated for expression of SMAD4, PAK3, p-JNK, and p-Jun by IHC. Therefore, different tumor specimens from different patients were evaluated. To directly show the clinical relevance of SMAD4, PAK3, p-JNK, and p-Jun, the set of paired samples taken from the same patient should be evaluated.
4. Lines 172-173. "Smad4" should be "loss of function of Smad4".
5. As stated in the lines 102-104, Smad4 is an important mediator of TGF-beta induced EMT. In this study (Fig 2), loss of function of Smad4 resulted in facilitating tumor cell motility, which is a typical phenotype of mesenchymal cells. How EMT markers, such as E-cadherin, SNAIL1 and TWIST1, were affected by loss of function of Smad4 in primary lung cancer cells used in Fig 2?
6. Tables were not labeled appropriately. Since the tables are too big to see, these should be moved to supplementary materials.

Reviewer #2 (Remarks to the Author):

I thank the authors for addressing all of my previous comments and congratulate them on a convincing study illustrating that SMAD4 regulates mir expression to control PAK3 and lung cancer metastasis.

I have a couple of remaining minor points.

It would be good to show all of the correlation graphs of the correlations shown in Figure 7C as you have done for SMAD4 and PAK3 expression in Figure S9D

Line 275 PK should read SPK

Reviewer #3 (Remarks to the Author):

The authors have done a lot of work to improve this ms, but there are still items that fall short:

The connection between Pak3 elevation and JNK activity is not clear. Despite some early reports that massive overexpression of Group A Paks can activate JNK, there is little evidence for a direct pathway (e.g., through a Jun kinase kinase) or a physiological role. In most of the non-transfected experiments here, the activation of JNK is modest at best. And anyway, even if there is a correlation between Pak3 levels and P-JNK, we don't know if it matters. To put this issue to rest, the authors could block JNK activity and show that this reversed the motility effects.

We still need evidence for actual Pak3 activation. The way to do this is either by P-Pak blots, IP-kinase assay for Pak3, and/or blotting for a direct target of Pak, like Mek P-S298

Reviewer #5 (Remarks to the Author):

The authors have adequately addressed the reviewer's comments.

Point-by-point Response to the Reviewers' Comments

Reviewer #1 (Remarks to the Author):

Authors examined the role of Smad4 (loss of function) in genetically engineered mouse models for lung cancer. They found that Smad4 loss of function with KRAS and TP53 (loss of function) mutations facilitated the production of metastasis. The metastasis was associated with activation of PAC3, which stimulated tumor cell motility, due to downregulation of miR-495 and miR-543. They further showed inverse correlation between Smad4 and PAK3/pJNK/pJun in clinical specimens obtained from lung cancer patients. The mechanism by which Smad4 loss of function resulted in increase of metastasis was clearly shown. However, there are several issues need to be addressed.

Major comments

1. How authors evaluated the production of metastasis in mouse models? The methods for detection of metastasis was not stated in the manuscript.

Response: Sorry for missing this description in the manuscript. We collected various mouse organs and processed them to get the paraffin blocks. Then we sectioned half of the organs (0.5 μ m thickness per slide) and selected five representative slides from each block to do H&E staining. Once there were tumors found in organs other than lungs, the immunohistochemical staining on TTF1 would be conducted. Given that TTF1 is a good immunohistochemical marker to distinguish a primary tumor from metastatic lung adenocarcinoma (PMID: 11156325), the tumor lesions with positive staining of TTF1 were considered as metastasized from the lungs.

Moreover, these metastasized lesions were from lung adenocarcinoma based on pathological morphology (Fig. 1B, 1D; Fig. S1B, S1E) and RNA-Seq analysis. Our RNA-Seq analyses of PK and SPK tumors demonstrate a similar expression pattern of TTF1 among all the tumor cells (Sup. Data 1). Moreover, the staining of lung squamous cell carcinoma markers, such as KRT5, SOX2, and Trp63, was undetected.

Below is this paragraph added in the Method section of the revised manuscript.

“To evaluate the production of metastasis in mouse models, half of the organ sample was sectioned into slides (0.5 μ m thickness per slide), and five representative slides were selected from each sample for H&E staining. Once there are tumors found in organs other than lungs, the immunohistochemical staining on TTF1 (a marker to demonstrate the origin of metastatic lesions from lung adenocarcinoma) would be conducted, and the tumors with the positive staining of TTF1 are considered as metastasized from the lungs.”

Part of Sup. Data 1 (Lung Adenocarcinoma marker: TTF1; Lung Squamous Cell Carcinoma marker: Krt5, SOX2, and Trp63)

gene_id	Expression_pk1	Expression_pk2	Expression_pk3	Expression_spk1	Expression_spk2	Expression_spk3
Ttf1	3.38809	3.40657	3.09721	3.68543	3.54401	3.77513
Krt5	0.306589	0.43149	0.684918	0	0	0
Sox2	0	0	0	0	0.427328	0.448323
Trp63	0	0	0	0.0134092	0.0195926	0.0138678

2. How authors evaluated the staining intensity in IHC. The definition of the staining intensity (+, +++, etc) was not stated in the manuscript.

Response: Sorry for missing the description in the Methods section for the evaluation of IHC staining intensity. We utilized Image-Pro to select cells with the positive staining by calculating the grey values. The combination of size ratio between the grey areas and the whole field as well as staining intensity were calculated, and we utilized the resulting index as our definition of the staining intensity (+, +++, etc.). The negative (-) means the percentage of the positive cells with less than 25% and near background staining; the index of the positive cells between 25% and 50% with low intensity is considered as weak (+); those between 50% and 75% with intermediate staining refers to moderate staining (++); those higher than 75% or more than 50% with strongest staining represents intense staining (+++).

Below is this paragraph added in the Method section of the revised manuscript.

“To evaluate the staining intensity in IHC, the white-view pictures on the slides were taken under the same condition. We utilized Image-Pro to select cells with the positive staining by calculating the grey values. The combination of size ratio between the grey areas and the whole field as well as staining intensity were calculated, and we utilized the resulting index as our definition of the staining intensity (+, +++, etc.). The negative (-) means the percentage of the positive cells with less than 25% and near background staining; the index of the positive cells between 25% and 50% with low intensity is considered as weak (+); those between 50% and 75% with intermediate staining refers to moderate staining (++); those higher than 75% or more than 50% with strongest staining represents intense staining (+++).”

3. Figure 7. 15 early, 12 advanced, and 30 metastatic lung cancer specimens were evaluated for expression of SMAD4, PAK3, p-JNK, and p-Jun by IHC. Therefore, different tumor specimens from different patients were evaluated. To directly show the clinical relevance of SMAD4, PAK3, p-JNK, and p-Jun, the set of paired samples taken from the same patient should be evaluated.

Response: Thank you so much! We fully agreed with the reviewer. And it is a great idea to evaluate the expression of SMAD4, PAK3, p-JNK, and p-Jun in the set of paired samples. Unfortunately, after consulting our collaborators in different hospitals,

we realized that samples obtained in this study were not paired. Therefore, we plan to do so it in our future studies. We sincerely appreciate that the review understands the difficulties we confronted.

4. Lines 172-173. “Smad4” should be “loss of function of Smad4”.

Response: Sorry for the mistake. We have revised it accordingly.

5. As stated in the lines102-104, Smad4 is an important mediator of TGF-beta induced EMT. In this study (Fig 2), loss of function of Smad4 resulted in facilitating tumor cell motility, which is a typical phenotype of mesenchymal cells. How EMT markers, such as E-cadherin, SNAIL1 and TWIST1, were affected by loss of function of Smad4 in primary lung cancer cells used in Fig 2?

Response: Our RNA-Seq results (Sup. Data 1) showed that there was no change in the expression of E-cadherin (CDH1) and TWIST1, while the expression of SNAIL1 (SNAIL) was significantly higher in SPK cells compared with PK cells. We further conducted Western blot analysis in PK and SPK cells (New data: Fig. S2G). The protein expression in E-cadherin and TWIST1 has no changes. Consistent with the change of SNAIL1 mRNA expression, we observed a relative increase of SNAIL1 protein expression in SPK cells compared with PK cells. The reason to observe the unchanged E-cadherin or TWIST1 may be that the regulation was not TGF β -dependent. For example, PAK3 expression was negatively regulated by SMAD4 in a TGF β -independent manner (Fig. S3A and S3H).

We also revised the manuscript text accordingly.

Fig. S2G. Western blot analysis of protein expression in PK and SPK cells.

6. Tables were not labeled appropriately. Since the tables are too big to see, these should be moved to supplementary materials.

Response: We have moved all the tables to supplementary materials as the Sup. Data.

Reviewer #2 (Remarks to the Author):

I thank the authors for addressing all of my previous comments and congratulate them on a convincing study illustrating that SMAD4 regulates mir expression to control PAK3 and lung cancer metastasis.

I have a couple of remaining minor points.

It would be good to show all of the correlation graphs of the correlations shown in Figure 7C as you have done for SMAD4 and PAK3 expression in Figure S9D

Response: According to your suggestions, we have added all of the correlation graphs of the correlations shown in Figure 7C as we have done for SMAD4 and PAK3 expression in Figure S9D. Please find the related data in Fig. S10 (New data).

Fig. S10. Pearson correlation analysis of Fig.7A-B.

Line 275 PK should read SPK

Response: Thanks a lot. We have revised it accordingly.

Reviewer #3 (Remarks to the Author):

The authors have done a lot of work to improve this ms, but there are still items that fall short:

The connection between Pak3 elevation and JNK activity is not clear. Despite some early reports that massive overexpression of Group A Paks can activate JNK, there is little evidence for a direct pathway (e.g., through a Jun kinase kinase) or a physiological role. In most of the non-transfected experiments here, the activation of JNK is modest at best. And anyway, even if there is a correlation between Pak3 levels and P-JNK, we don't know if it matters. To put this issue to rest, the authors could block JNK activity and show that this reversed the motility effects.

Response: According to the reviewer's suggestion, we blocked JNK activity with SP600125, an inhibitor of P-JNK, and examined the effect on cell motility. While vehicle treated SPK cells migrated faster than PK cells, there was no difference in cell motility after the SP600125 treatment (New data: Fig. S4H), suggesting that blocking JNK activity abolished the impact of PAK3 on cell motility.

We also revised the manuscript text accordingly.

Fig. S4H. Migration assay of PK and SPK cells.

We still need evidence for actual Pak3 activation. The way to do this is either by P-Pak blots, IP-kinase assay for Pak3, and/or blotting for a direct target of Pak, like Mek P-S298

Response: As suggested by the Reviewer, we examined the expression of MEK P-S298. We found that P-MEK was relatively stronger in the SPK cells compared with that of PK cells (New data: Fig. S4I). It indicated actual PAK3 activation in SPK cells.

We also revised the manuscript text accordingly.

Fig. S4I. Western blot analysis of protein expression in PK and SPK cells.

Reviewer #5 (Remarks to the Author):

The authors have adequately addressed the reviewer's comments.

Response: Thanks a lot.

REVIEWERS' COMMENTS

Reviewer #1 (Remarks to the Author):

Authors addressed the majority of the reviewer's comments.

Major comment 1.

Authors stated that "we sectioned half of the organs (0.5um thickness). Is the thickness really 0.5um, not 5um?"

Reviewer #3 (Remarks to the Author):

I am satisfied with the revisions and have no remaining issues.